# Protein Autoregressive Modeling via Multiscale Structure Generation

Yanru Qu [* 1 2]   Cheng-Yen Hsieh [* † 1]   Zaixiang Zheng [1]   Ge Liu [2]   Quanquan Gu [1]

## Abstract

We present *protein autoregressive modeling* (PAR), the *first* multi-scale autoregressive framework for protein backbone generation via coarse-to-fine next-scale prediction. Using the hierarchical nature of proteins, PAR generates structures that mimic sculpting a statue, forming a coarse topology and refining structural details over scales. To achieve this, PAR consists of three key components: *(i)* multi-scale downsampling operations that represent protein structures across multiple scales during training; *(ii)* an autoregressive transformer that encodes multi-scale information and produces conditional embeddings to guide structure generation; *(iii)* a flow-based backbone decoder that generates backbone atoms conditioned on these embeddings. Moreover, autoregressive models suffer from exposure bias, caused by the training and the generation procedure mismatch, and substantially degrades structure generation quality. We effectively alleviate this issue by adopting noisy context learning and scheduled sampling, enabling robust backbone generation. Notably, PAR exhibits strong zero-shot generalization, supporting flexible human-prompted conditional generation and motif scaffolding *without* requiring fine-tuning. On the unconditional generation benchmark, PAR effectively learns protein distributions and produces backbones of high design quality, and exhibits favorable scaling behavior. Together, these properties establish PAR as a promising framework for protein structure generation. Project page: https://par-protein.github.io

*Equal contribution †Project Lead. This work was done during Yanru Qu's internship. [1]ByteDance Seed [2]University of Illinois at Urbana-Champaign. Correspondence to: Quanquan Gu <quanquan.gu@bytedance.com>.

*Proceedings of the 43rd International Conference on Machine Learning*, Seoul, South Korea. PMLR 306, 2026. Copyright 2026 by the author(s).

## 1. Introduction

Deep generative modeling of proteins has emerged as a way to design and model novel structures with desired functions and properties, with broad applications in biomedicine and nanotechnology (Huang et al., 2016; Kuhlman & Bradley, 2019). A widely adopted approach is to directly model the distribution of three-dimensional protein structures, which govern protein function. Typically, structure generative models produce protein backbones without sequences or side chains. Prior work in this area could be broadly categorized into methods that predict the SE(3) backbone frame representations (Yim et al., 2023a; Watson et al., 2023) and those that directly model atoms, *e.g.*, C$\alpha$ coordinates for simplicity and scalability (Geffner et al., 2025; Lin & AlQuraishi, 2023). However, all these works are based on diffusion models and their variations (*e.g.*, flow matching).

On the other hand, autoregressive (AR) modeling has emerged as a powerful paradigm for large language models (Achiam et al., 2023; Touvron et al., 2023). AR models employ *next-token prediction* to model the probability of each token based on prior ones, showing striking empirical behaviors such as scalability (Kaplan et al., 2020) and *zero-shot generalization* to unseen tasks (Brown et al., 2020).

Despite its success in other domains, AR modeling has received little attention in backbone modeling. We identify two main reasons. **(i)** Extending AR models to continuous data, *e.g.* atomic positions in 3D, often relies on data discretization (Esser et al., 2021b;a), which can reduce structural fidelity and fine-grained details for proteins, limiting generative performance (Hsieh et al., 2025). **(ii)** Protein residues exhibit strong *bidirectional* dependencies: residues distant in sequence may be spatially close and form hydrogen bonds or hydrophobic contacts. This mutual dependency conflicts with the *unidirectional* assumption of standard AR models, and thus limits the quality of previous attempts on autoregressive structure generation (Gaujac et al., 2024). A natural question therefore arises: *can we apply AR modeling to protein backbone design?*

In this paper, we answer the above question affirmatively, and propose **PAR**, a **p**rotein **a**uto**r**egressive framework, to unlock the power of AR models for protein backbone generation. We take initiative from the hierarchical nature of proteins: their structures span multiple *scales* of granu-

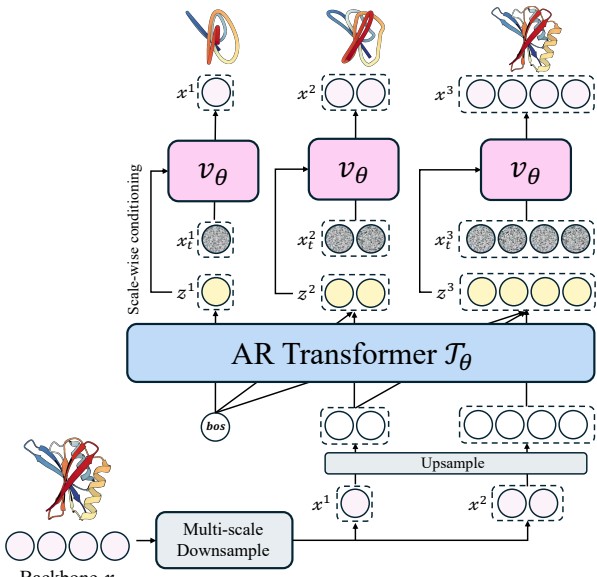

Figure 1. **Overview of PAR.** PAR comprises the autoregressive (AR) transformer $\mathcal{T}_\theta$ and the flow-based backbone decoder $\mathbf{v}_\theta$. During training, we downsample a backbone $\mathbf{x} \in \mathbb{R}^{L \times 3}$ into multi-scale representations $\{\mathbf{x}^1, \ldots, \mathbf{x}\}$. *AR transformer* performs next-scale prediction, producing conditional embeddings $(\mathbf{z}^1, \ldots, \mathbf{z}^n)$ from $(bos, \ldots, \mathbf{x}^{n-1})$. The shared *flow-based decoder* learns to denoise backbones $\mathbf{x}^i$ at each scale conditioned on $\mathbf{z}^i$. At inference, PAR autoregressively generates $\mathbf{x}^i$ until the final structure $\mathbf{x}$ is constructed.

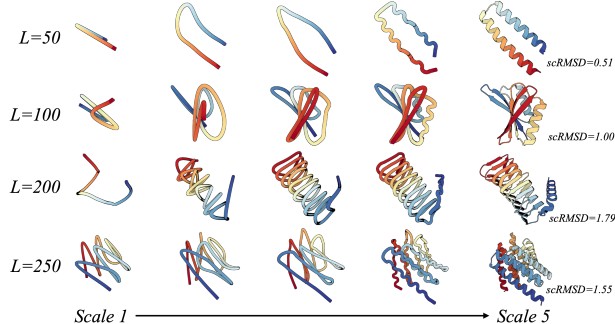

Figure 2. **Samples generated by PAR over scales.** We illustrate PAR's generation process across five scales. Much like sculpting a statue, the model first formulates the global structural layout at coarse scales and progressively refines the details at later scales.

larity, from coarse 3D topology and tertiary fold arrangements, local secondary structures, to the finest atomic coordinates. PAR thus adopts a multi-scale autoregressive framework via next-scale prediction, predicting each scale conditioned on prior coarser scales. This strategy, inspired by advances in image generation, enabled AR models to surpass strong diffusion models in image synthesis for the first time (Tian et al., 2024), and further allows multimodal LLMs to achieve unified text and image generation framework (Li et al., 2025).

Building on this multi-scale framework, PAR includes three key components (Fig. 1). The *multi-scale downsampling* creates coarse-to-fine structural representations to serve as structural context and targets during training. *AR transformer*, a stack of non-equivariant attention layers (Vaswani et al., 2017), encodes all preceding scales to produce a scale-wise conditional embedding following Li et al. (2024). The *flow-based backbone decoder* is conditioned on this embedding to model C$\alpha$ backbone atoms directly. As a result, PAR avoids both discretization of protein structures and residue-wise unidirectional autoregressive ordering, thereby overcoming the two aforementioned limitations that compromise structural fidelity and generative quality. Moreover, training on ground-truth structural context, AR models suffer from *exposure bias* (Arora et al., 2022), which is a key challenge substantially reducing structure generation quality

in our preliminary study. We effectively mitigate such issue via noisy context learning and scheduled sampling, allowing the model to learn from partially corrupted context.

This multi-scale approach introduces several notable model behaviors. PAR generates backbones by establishing a global topology and performing refinements, analogous to progressively sculpting a statue into a masterpiece (Fig. 2). For unconditional generation, PAR exhibits favorable scaling behavior, yielding competitive results on distributional metrics like Fréchet Protein Structure Distance (FPSD). Unlike diffusion models, which operate at a single scale, PAR flexibly handles inputs at various granularities, and hence shows zero-shot generalization in tasks like prompt-based generation and motif scaffolding. The multi-scale formulation enables PAR to orchestrate sampling strategies, achieving a **2.5x** sampling speedup compared to single-scale baselines. Finally, PAR provides a more general framework, incorporating flow-based models as a special case when restricted to a single scale, and thus remains compatible with techniques from flow-based models like self-conditioning (Chen et al., 2023).

**Main contributions:** *(i)* We present PAR, the *first* multi-scale AR model for protein backbone generation that addresses key limitations of existing AR methods. *(ii)* PAR comprises multi-scale downsampling, AR transformer, and a flow-based decoder, to directly model C$\alpha$ atom, avoiding discretization loss. *(iii)* We alleviate exposure bias through noisy context learning and scheduled sampling, effectively improving structure generation. *(iv)* Our model shows an interpretable generation process that forms coarse backbone topology and refines it progressively. *(v)* Benchmarking results show that PAR effectively captures protein data distributions, achieving FPSD score of **161.0** against PDB dataset that further scale with training compute. *(vi)* PAR exhibits efficient sampling and zero-shot generalization potential, reflecting the versatility of AR large language models.

**Conflict of Interest Disclosure.** Most authors are employed by ByteDance, where the model described in this work was developed as part of research activities.

## 2. Background and Related Work

**Flow and diffusion-based structure generative models.** Flow-based and diffusion methods (Lipman et al., 2022; Ho et al., 2020) operate by transforming samples from a prior distribution to the target data distribution, and have been widely applied to protein backbone generation. These methods either predict per-residue rotations and translations using a frame-based Riemannian manifold representation (Yim et al., 2023b; Bose et al., 2024; Yim et al., 2023a; Watson et al., 2023; Ingraham et al., 2023) or directly model atom coordinates, such as $C\alpha$ positions (Lin & AlQuraishi, 2023; Lin et al., 2024; Geffner et al., 2025), with some approaches generating fully atomistic proteins including side chains (Qu et al., 2025; Chu et al., 2024). Discrete diffusion methods (Hayes et al., 2025; Wang et al., 2025), trained on structure tokens, often reduce structural fidelity and limit generation quality (Hsieh et al., 2025). Unlike most diffusion approaches, which are single-scale, PAR models protein structures across multiple scales using a parameterized upsampling autoregressive process from short to long, allowing flexible handling of different structural granularities and zero-shot generalization to tasks like prompt-based generation. In addition, PAR provides a more general framework, as it naturally reduces to a flow-based model when restricted to a single scale.

**Autoregressive modeling.** Autoregressive (AR) modeling has been driving natural language processing and computer vision due to its strong scalability and zero-shot generalization (Tian et al., 2024; Touvron et al., 2023; Achiam et al., 2023). The approach relies on *next-token prediction* that predicts the distribution of the next token based on prior ones in a *unidirectional* sequence. However, adapting autoregressive models to continuous domains, like image generation, often involves tokenizers such as VQVAE (Esser et al., 2021b;a), which discretizes the data for transformer training and may discard fine-grained details. Recently, Li et al. (2024) used the AR model that produces conditioning for a diffusion network (*e.g.*, a small MLP) to model image latents, unlocking the operations of AR models in a continuous-valued space. In addition, defining appropriate autoregressive orders that preserve data properties is crucial. Since next-token prediction inherently discards spatial locality by flattening the 2D image feature map into a 1D sequence, VAR (Tian et al., 2024) introduced *next-scale prediction*. Leveraging a multi-scale VQVAE, the image feature map is quantized into $n$ multi-scale token maps that preserve the spatial and *bidirectional* correlations. To our knowledge, autoregressive modeling has not been widely applied to protein structure generation despite their success in other domains. The only exception is Gaujac et al. (2024), which models structure tokens with a causal transformer. In contrast, we design a multi-scale autoregressive framework that operates directly in continuous backbone space using a flow-based backbone decoder, thereby addressing the limitations of discrete token maps while respecting the bidirectional biophysical relations of protein structures.

## 3. Protein Autoregressive Modeling

In this section, we introduce PAR, a multi-scale autoregressive (AR) framework for protein backbone generation. Formally, we want to model a protein backbone $C\alpha$ structure with $L$ residues $\mathbf{x} \in \mathbb{R}^{L \times 3}$ in an autoregressive manner as follows:

$$
\begin{aligned}
p_\theta(\mathbf{x}) &= \mathbb{E}_{X \sim q_{\text{decompose}}(\cdot|\mathbf{x})} \left[ p_\theta(X = \{\mathbf{x}^1, \ldots, \mathbf{x}^n\}) \right] \\
&= \mathbb{E}_{X \sim q_{\text{decompose}}(\cdot|\mathbf{x})} \prod_{i=1}^{n} p_\theta(\mathbf{x}^i \mid X^{<i}).
\end{aligned}
\tag{1}
$$

where $q_{\text{decompose}}(\cdot|\mathbf{x})$ defines a decomposition of autoregressive order for protein structure $\mathbf{x}$ into $n$ scales $X = \{\mathbf{x}^1, \ldots, \mathbf{x}^n\}$ with $\mathbf{x}^n = \mathbf{x}$, while $p_\theta(\mathbf{x}^i \mid X^{<i})$ is the desired PAR model learning to generate $\mathbf{x}$ via a scale-wise autoregression.

The design space of $q_{\text{decompose}}$ and $p_\theta$ under this formulation (Eqn. 1) can be flexible. Recall that our goal is to enable AR modeling to preserve spatial dependencies and avoid discretization, as discussed in §1. To this end, in §3.1, we devise a non-parametric and deterministic $q_{\text{decompose}}$ by *multi-scale protein downsampling* (Fig. 1) that represents protein backbones at multiple scales via hierarchical downsampling (Eqn. 2), providing structural context and training targets. In §3.2, we parameterize PAR $p_\theta$ as a backbone autoregressive upsampling process via next-scale prediction and achieve direct $C\alpha$ modeling in the continuous space (Eqn. 3). This comprises two key components: *(i)* an autoregressive transformer (Fig. 1) that produces scale-wise conditional embeddings informed by preceding scales to guide generation (Eqn. 4); and *(ii)* a flow-based backbone decoder (Fig. 1) which samples $C\alpha$ backbone coordinates conditioned on the learned embeddings (Eqn. 5).

Finally, in §3.3, we dedicated strategies to mitigate *exposure bias* (Arora et al., 2022), a mismatch between training on ground-truth data and inference on model predictions that leads to error accumulations and degrading generation quality in AR models. Together, these components enable PAR to robustly generate protein backbones in a coarse-to-fine manner.

### 3.1. Multi-Scale Protein Downsampling

We construct the multi-scale representations of protein structures via hierarchical downsampling to serve as training context and targets for PAR (Fig. 1). Given a protein structure $\mathbf{x} \in \mathbb{R}^{L \times 3}$, it produces a hierarchy of coarse-to-fine scales by progressively downsampling $\mathbf{x}$ into $n$ scales:

$$
\begin{aligned}
q_{\text{decompose}} : \ \mathbf{x} &\mapsto X = \{\mathbf{x}^1, \mathbf{x}^2, \ldots, \mathbf{x}^n\} \\
&= \{\text{Down}(\mathbf{x}, \texttt{size}(1)), \text{Down}(\mathbf{x}, \texttt{size}(2)), \ldots, \mathbf{x}\}.
\end{aligned}
\tag{2}
$$

where $\text{Down}(\mathbf{x}, \texttt{size}(i)) \in \mathbb{R}^{\texttt{size}(i) \times 3}$ denotes a downsampling operation that interpolates $\mathbf{x}$ along the sequence dimension, leading to $\texttt{size}(i)$ 3D centroids that provide a coarse structural layout. Since $q_{\text{decompose}}$ is designed as a deterministic mapping for every $\mathbf{x}$, the likelihood of Eqn. 1 can be simplified without marginalization: $p_\theta(\mathbf{x}) = \prod_{i=1}^n p_\theta(\mathbf{x}^i \mid X^{<i})$. We show that this downsampling strategy properly preserves pairwise spatial relationships in §C.8.

**Scale configurations** $\mathcal{S} = \{\texttt{size}(1), \ldots, \texttt{size}(n)\}$ could be defined in two ways. When defined *by length,* scales are chosen as hyperparameters, *e.g.*, $\mathcal{S} = \{64, 128, 256\}$. In this case, if $L$ lies in $(\texttt{size}(i), \texttt{size}(i+1)]$, the protein could be generated with only $i + 1$ autoregressive steps. When defined *by ratio,* scales are adaptively determined based on protein length, *e.g.*, $\mathcal{S} = \{L/4, L/2, L\}$. Empirically, defining scales by length yields slightly better results in modeling data distributions. We adopt this as the default configuration. This design enables training PAR with flexible scale configurations. In the following sections, we describe how this hierarchy of representations are modeled using the autoregressive transformer and backbone decoder.

### 3.2. Coarse-to-Fine Backbone Autoregressive Modeling

Preserving the inherent dependencies in data when defining the autoregressive order is crucial and affects generation performance (Tian et al., 2024). Standard AR models assume *unidirectional* dependency, which conflicts with the strong *bidirectional* interactions in protein sequences, *e.g.*, spatially close residues can form hydrophobic contacts or hydrogen bonds even if distant in sequence. PAR addresses this with a multi-scale AR framework via next-scale prediction, capturing mutual structural dependency over each scale. Motivated by Li et al. (2024), we propose to use an AR Transformer with diffusion/flow-based regression loss to enable modeling of C$\alpha$ atoms directly in continuous space. That is, we could rewrite the likelihood as:

$$
\begin{aligned}
p_\theta(X = \{\mathbf{x}^1, \ldots, \mathbf{x}^n\}) &= \prod_{i=1}^n p_\theta(\mathbf{x}^i | X^{<i}) \\
&= \prod_{i=1}^n p_\theta(\mathbf{x}^i \mid \mathbf{z}^i = \mathcal{T}_\theta(X^{<i})),
\end{aligned}
\tag{3}
$$

where $\mathcal{T}_\theta$ is an AR Transformer that produces scale-wise conditioning $\mathbf{z}^i$ while $p_\theta(\mathbf{x}^i|\mathbf{z}^i)$ is optimized with a flow-based atomic decoder $\mathbf{v}_\theta$ with flow matching. This avoids discretizing protein structures into tokens, preserving structural details and generation fidelity. We describe each component below.

**Autoregressive transformer for scale-wise conditioning.** To formulate the autoregressive order, we leverage the hierarchical nature of proteins, where a protein structure could span various levels of representations from coarse tertiary topology to the finest atomic coordinates. We adopt the next-scale prediction to model per-scale distribution based on prior coarser scales, which further ensures that the *bidirectional* dependencies of residues are modeled over each scale. We train our autoregressive model (Fig. 1), a non-equivariant transformer $\mathcal{T}_\theta$, to produce scale-wise conditioning embedding $\mathbf{z}^i$ for scale $i$ depending on prior scales $X^{<i} = \{\mathbf{x}^1, \ldots, \mathbf{x}^{i-1}\}$:

$$
\begin{aligned}
\mathbf{z}^i = \mathcal{T}_\theta(X^{<i}) = \mathcal{T}_\theta\Big( \big[\texttt{bos}, \text{Up}(\mathbf{x}^1, \texttt{size}(2)), \\
\ldots, \text{Up}(\mathbf{x}^{i-1}, \texttt{size}(i))\big]\Big).
\end{aligned}
\tag{4}
$$

where $\texttt{bos} \in \mathbb{R}^{\texttt{size}(1) \times 3}$ is a learnable embedding, and $\text{Up}(\mathbf{x}^{i-1}, \texttt{size}(i))$ interpolates $\mathbf{x}^{i-1}$ to $\texttt{size}(i)$ 3D points. All inputs are concatenated along the sequence dimension before being fed into $\mathcal{T}_\theta$. The embedding $\mathbf{z}^i$ is then used to condition the flow matching decoder to predict the backbone coordinates $\mathbf{x}^i$, detailed as follows.

**Flow-based atomic decoder.** We enable PAR to directly model C$\alpha$ positions $\mathbf{x}$, wherein $p_\theta(\mathbf{x}|\mathbf{z}^i)$ is parameterized by an atomic decoder $\mathbf{v}_\theta$ with flow matching (FM, Lipman et al., 2022), which maps standard normal distribution to the target data distribution. We condition the $\mathbf{v}_\theta$ with scale-wise conditioning $\mathbf{z}^i$ predicted by the AR Transformer $\mathcal{T}_\theta$ at each scale $i$ (Fig. 1). During training, we sample the noise $\boldsymbol{\epsilon}^i \sim \mathcal{N}(0, I)$ and a time variable $t^i \in [0, 1]$, and compute the interpolated sample as $\mathbf{x}_{t^i}^i = t^i \cdot \mathbf{x}^i + (1 - t^i) \cdot \boldsymbol{\epsilon}^i$. As such, we can jointly train $\mathbf{v}_\theta$ and $\mathcal{T}_\theta$ with an FM objective:

$$
\begin{aligned}
\mathcal{L}(\theta) = \mathbb{E}_{\mathbf{x} \sim p_\mathcal{D}} \Bigg[ \frac{1}{n} \sum_{i=1}^n \frac{1}{\texttt{size}(i)} \mathbb{E}_{t^i \sim p(t^i), \boldsymbol{\epsilon}^i \sim \mathcal{N}(0, I)} \\
\big\| \mathbf{v}_\theta(\mathbf{x}_{t^i}^i, t^i, \mathbf{z}^i) - (\mathbf{x}^i - \boldsymbol{\epsilon}^i) \big\|^2 \Bigg].
\end{aligned}
\tag{5}
$$

where $p_\mathcal{D}(\mathbf{x})$ denotes the training data distribution and $p(t)$ denotes the t-sampling distribution in Geffner et al. (2025). The conditioning embedding $\mathbf{z}^i$ is injected into the atomic decoder network $\mathbf{v}_\theta$ through adaptive layer norms (Peebles & Xie, 2023). We further concatenate a learnable scale embedding alongside $\mathbf{z}^i$ to help the model identify different scales and incorporate self-conditioning input as an additional condition (Chen et al., 2023), though we omit them in the equation for simplicity. To formulate

*Table 1.* **Unconditional backbone generation performance**. We follow Geffner et al. (2025) in adopting **FPSD** and **fS** to evaluate the model's ability to capture the data distribution. $\mathrm{PAR}_{\mathrm{pdb}}$ denotes the 400M model finetuned on the PDB subset.

| Method | Designability | | FPSD vs. | | fS | Diversity | Novelty | Sec. Struct. % |
| | (%)↑ | sc-RMSD↓ | PDB↓ | AFDB↓ | (C / A / T)↑ | TM-Sc.↓ | TM-Sc.↓ | ($\alpha/\beta$) |
|---|---|---|---|---|---|---|---|---|
| FrameDiff (17M) | 65.4 | - | 194.2 | 258.1 | 2.46/5.78/23.35 | 0.40 | 0.82 | 64.9/11.2 |
| RFDiffusion (60M) | 94.4 | - | 253.7 | 252.4 | 2.25/5.06/19.83 | 0.42 | 0.85 | 64.3/17.2 |
| ESM3 (1.4B) | 22.0 | - | 933.9 | 855.4 | **3.19**/6.71/17.73 | 0.42 | 0.90 | 64.5/8.5 |
| Genie2 (16M) | 95.2 | - | 350.0 | 313.8 | 1.55/3.66/11.65 | 0.38 | 0.80 | 72.7/4.8 |
| Proteina (200M) | 92.8 | 1.14 | 282.3 | 285.6 | 2.17/6.22/21.48 | 0.37 | 0.85 | 66.3/9.2 |
| Proteina (400M) | 92.6 | 1.09 | 271.3 | 272.6 | 2.13/6.14/21.18 | 0.37 | 0.84 | 65.1/9.5 |
| PAR (200M) | 87.0 | 1.33 | 252.0 | 237.9 | 2.11/6.41/19.22 | 0.37 | 0.83 | 64.3/8.8 |
| PAR (400M) | 96.0 | **1.01** | 313.9 | 296.4 | 2.24/6.60/16.71 | 0.39 | 0.85 | 66.3/8.9 |
| $\gamma$=0.45 | 88.0 | 1.28 | 231.5 | **211.8** | 2.20/6.59/20.96 | **0.36** | 0.84 | 63.2/9.7 |
| $\mathrm{PAR}_{\mathrm{pdb}}$ | **96.6** | 1.04 | **161.0** | 228.4 | 2.57/7.42/23.61 | 0.43 | 0.85 | 50.2/16.7 |
| $\gamma$=0.45 | 88.8 | 1.34 | 176.6 | 256.4 | 2.62/**7.52/30.99** | 0.40 | 0.84 | 50.2/16.2 |

the indices for positional encoding $p^i$ at scale $i$, we uniformly sample $\mathtt{size}(i)$ numbers from the interval $[1, L]$, *i.e.*, $p^i = \mathrm{linspace}(1, L, \mathtt{size}(i))$. At coarse scales, the wide spacing between adjacent indices encourages the model to capture global structural layout, while at finer scales the dense indices allow the model to focus on local details. For more details, please refer to §A.1.

Leveraging the learned flow network $\mathbf{v}_\theta$, sampling could be performed at each scale through ordinary differential equation (ODE) $d\mathbf{x}_t = \mathbf{v}_\theta(\mathbf{x}_t, t) \, dt$, with the scale superscript $i$ and condition $\mathbf{z}$ omitted for simplicity. Moreover, we could define the stochastic differential equation (SDE) for sampling:

$$d\mathbf{x}_t = \mathbf{v}_\theta(\mathbf{x}_t, t) \, dt + g(t) \, \mathbf{s}_\theta(\mathbf{x}_t, t) \, dt + \sqrt{2g(t)\gamma} \, d\mathcal{W}_t, \tag{6}$$

where $g(t)$ is a time-dependent scaling function for the score function $\mathbf{s}_\theta(\mathbf{x}_t, t)$ (Albergo et al., 2025; Ma et al., 2024) and the noise term, $\gamma$ is a noise scaling parameter, and $\mathcal{W}_t$ is a standard Wiener process. The score function, defined as the gradient of the log-probability of the noisy data distribution at time $t$, could be computed as $\mathbf{s}_\theta(\mathbf{x}_t, t) = \frac{t \, \mathbf{v}_\theta(\mathbf{x}_t, t) - \mathbf{x}_t}{1 - t}$.

**Multi-scale structure generation.** At inference, the autoregressive transformer first produces $\mathbf{z}^1$ at the coarsest scale, which conditions the flow matching decoder to generate $\mathbf{x}^1$ either via ODE or SDE sampling in Eqn. 6. We upsample $\mathbf{x}^1$ using $\mathrm{Up}(\mathbf{x}^1, \mathtt{size}(2))$ and send it back into the autoregressive transformer to predict the next scale embedding $\mathbf{z}^2$. This coarse-to-fine process iterates $n$ times until the flow-matching model generates the full-resolution backbone $\mathbf{x}$. KV cache is applied throughout the autoregressive process for efficiency.

### 3.3. Mitigating Exposure Bias

Training AR models typically uses teacher forcing (Williams & Zipser, 1989), where ground-truth data are fed as context to stabilize learning. However, during in-

ference the model is conditioned on its own predictions, creating a training-inference mismatch known as *exposure bias* (Arora et al., 2022; He et al., 2021). Errors can then accumulate across autoregressive steps, degrading output quality. Our preliminary study shows that teacher forcing greatly reduces the designability of generated structures. To mitigate this, we adapt Noisy Context Learning (NCL) and Scheduled Sampling (SS), techniques from language and image AR modeling (Ren et al., 2025; Bengio et al., 2015), for PAR.

**Noisy context learning.** We train PAR with noisy context, adding noise to the ground-truth prior-scale input during training. This encourages the model to learn the per-scale distribution without relying on perfectly accurate context, improving robustness. We randomly sample $n$ noise weights $\{w_{\mathrm{ncl}}^1, \cdots, w_{\mathrm{ncl}}^n\} \in [0, 1]$, and draw n noise samples $\{\boldsymbol{\epsilon}_{\mathrm{ncl}}^1, \cdots, \boldsymbol{\epsilon}_{\mathrm{ncl}}^n\} \in \mathcal{N}(0, I)$. Each input context $\mathbf{x}^i$ is corrupted as $\mathbf{x}_{\mathrm{ncl}}^i = w_{\mathrm{ncl}}^i \cdot \mathbf{x}^i + (1 - w_{\mathrm{ncl}}^i) \cdot \boldsymbol{\epsilon}_{\mathrm{ncl}}^i$. This perturbation is applied to the input context *only* during training, which updates the autoregressive step in Eqn. 4 as $\mathbf{z}^i = \mathcal{T}_\theta \Big( \big[ \mathtt{bos}, \mathrm{Up}(\mathbf{x}_{\mathrm{ncl}}^1, \mathtt{size}(2)), \ldots, \mathrm{Up}(\mathbf{x}_{\mathrm{ncl}}^{i-1}, \mathtt{size}(i)) \big] \Big)$.

**Scheduled sampling.** During training, we use scheduled sampling (Bengio et al., 2015) by running the forward process iteratively across scales. At the $i$-th scale, the flow-based backbone decoder predicts the clean data $\mathbf{x}_{\mathrm{pred}}^i = \mathbf{x}_t^i + (1 - t^i)\mathbf{v}_\theta(\mathbf{x}_t^i, t^i, \mathbf{z}^i)$. With a probability of 0.5, we replace the ground truth context $\mathbf{x}^i$ with this prediction $\mathbf{x}_{\mathrm{pred}}^i$ at later scales. This exposes the model to its own output and reduces the train-test gap. Notably, we could combine noisy context learning with this technique by adding noises to the model predicted context $\mathbf{x}_{\mathrm{pred}}^i$.

## 4. Experiments

We begin by evaluating PAR on unconditional backbone generation and compare it with existing structure generative methods in §4.1. Next, we examine its zero-shot general-

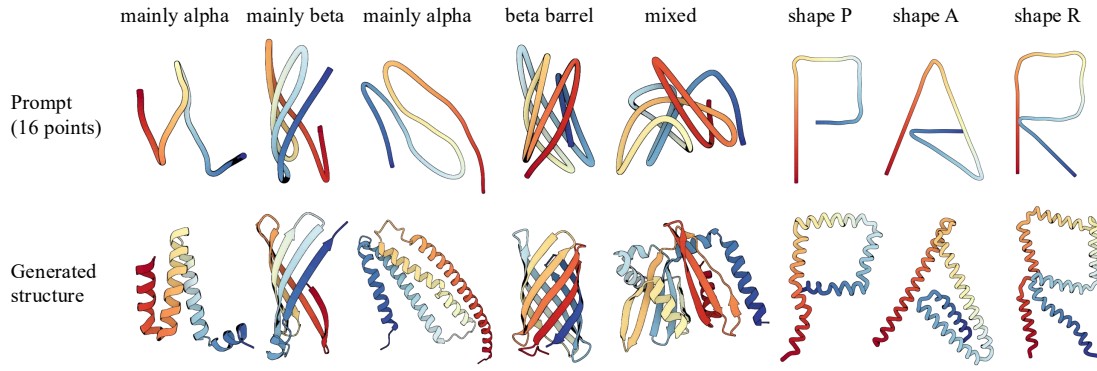

*Figure 3.* **Backbone generation with human prompt.** Given a small number of points (*e.g.*, 16) as prompt, PAR can generate protein backbones that adhere to the global arrangements specified by these points, *without* any finetuning. For visualization, input points are interpolated to match the length of the generated structure.

ization ability in §4.2. We then study the scaling behavior, efficient sampling, and propose strategies to mitigate exposure bias, along with additional ablations in §4.3. We include additional empirical analysis in §C.

### 4.1. Protein Backbone Generation

**Generation over scales.** We illustrate PAR's backbone generation using a 5-scale model ($\mathcal{S} = \{L/16, L/8, L/4, L/2, L\}$) in Fig. 2, showing generated structures with target lengths of $\{50, 100, 200, 250\}$ residues. Generation proceeds in a coarse-to-fine manner, which resonates with statue sculpting: the coarser scales establish a rough global layout, and finer scales progressively add local details. This multi-scale formulation yields a clear and interpretable generation process. We present the quantitative analysis on PAR's backbone generation in the next paragraph.

**Unconditional generation benchmark.** We compare 3-scale PAR's ($\mathcal{S} = \{64, 128, 256\}$) backbone generation performance with other baselines in Tab. 1, following the evaluation protocol in Geffner et al. (2025). The baselines span three categories: frame-based diffusion methods (Yim et al., 2023c; Watson et al., 2023), multimodal protein language models (Hayes et al., 2025), and diffusion/flow-based C$\alpha$ generators (Lin et al., 2024; Geffner et al., 2025). We disable the optional pair representations and triangle-based modules (Jumper et al., 2021) in both PAR and Proteina to align architectural capacity and improve computational efficiency. We train PAR using a two-stage procedure following Geffner et al. (2025): the model is first trained for 200K steps on the AFDB representative dataset and subsequently fine-tuned for 5K steps on a PDB subset of 21K designable samples. Results for the remaining baselines are taken directly from Geffner et al. (2025). Evaluation metrics and baseline categories are detailed in §A.2§A.3. To better reflect the goal of unconditional protein generation as modeling the full data distribution, we adopt FPSD,

which jointly measures quality and diversity by comparing generated and reference distributions, analogous to FID in image generation (Heusel et al., 2017). As shown in Tab. 1, PAR generates samples that closely match the reference data distribution and maintains competitive designability. On FPSD, PAR achieves scores of 211.8 against AFDB and 231.5 against PDB. By reducing the noise scaling parameter $\gamma$ (Equation 6) from 0.45 to 0.3 in SDE sampling, we can reduce sampling stochasticity and further improve sample quality, improving the designability from 88.0% to 96.00%. After fine-tuning, PAR achieved 96.6% designability and 161.0 FPSD against the PDB, highlighting its superior distributional fidelity compared to pure diffusion-based baselines. We provide additional analysis on longer proteins in §C.3.

### 4.2. Zero-Shot Task Generalization

**Guiding backbone generation with human-specified prompt.** Proteins possess hierarchical and complex structures, which makes it challenging to directly specify a target shape and design proteins accordingly. By leveraging PAR's coarse-to-fine generation, a simple prompt (e.g., 16 points) can specify a protein's coarse layout, from which the model generates the complete structure as shown in Fig. 3. In particular, we first obtain a 16-point input prompt either by downsampling a real protein structure from the test set, or by specifying the points manually (the top row in Fig. 3). Using a 5-scale PAR ($\mathcal{S} = \{16, 32, 64, 128, 256\}$), we initialize the first-scale prediction with the 16-point prompt and autoregressively upsample until the full protein structure is generated, as illustrated in the bottom row of Fig. 3. Following this process, PAR can generate a new structure that preserves the coarse structural layout (first five examples), and explore entirely novel structures (last three examples). If desired, longer prompts (*e.g.*, 32 points) could be specified to achieve more finer-grained control over backbone generation. As later shown in Tab. 5, we quantitatively evaluate the structural consistency (TM-score) between the

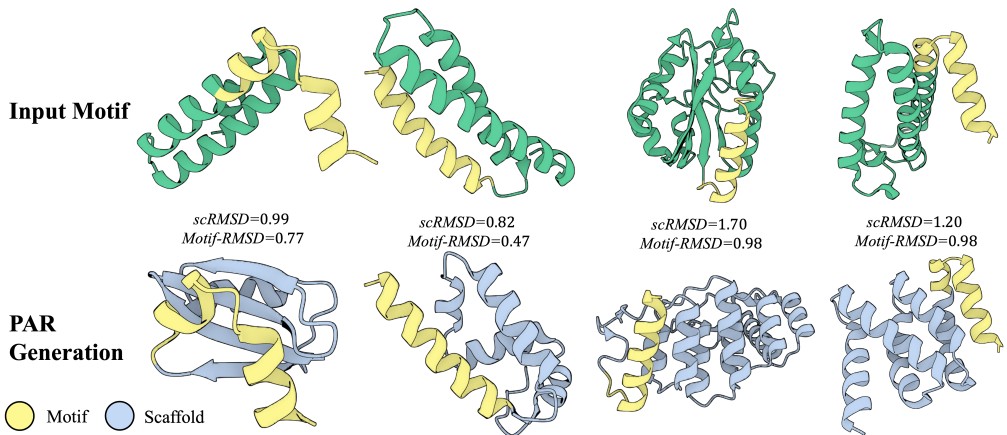

scRMSD=0.99          scRMSD=0.82          scRMSD=1.70          scRMSD=1.20
Motif-RMSD=0.77      Motif-RMSD=0.47      Motif-RMSD=0.98      Motif-RMSD=0.98

*Figure 4.* **Zero-shot motif scaffolding.** Given a motif structure, PAR can generate diverse, plausible scaffold structures that accurately preserve the motif via teacher-forcing the motif coordinates at each scale, without additional conditioning or fine-tuning.

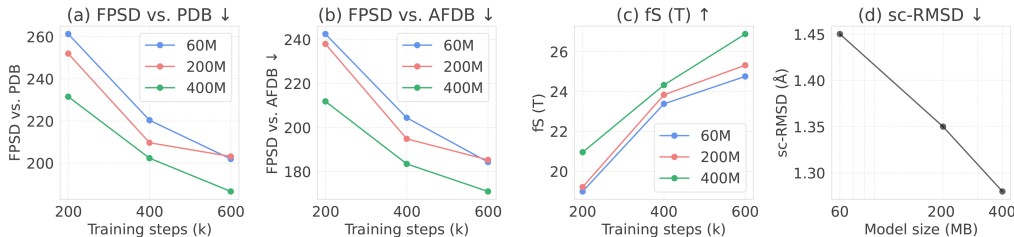

*Figure 5.* **Scaling effects of PAR.** Performance of four metrics over varying training steps and model sizes, (a) FPSD vs. PDB, (b) FPSD vs. AFDB, (c) fS(T), (d) sc-RMSD.

prompted layout and the final generation.

**Motif scaffolding.** Besides the point-based layout, PAR can preserve finer-grained prompts like atomic coordinates. Fig. 4 highlights the zero-shot motif scaffolding capabilities of PAR. Using a 5-scale PAR, we downsample a raw protein structure into five scales and teacher-force the ground-truth motif coordinates at each scale before propagating into the next scale. To avoid clashes or discontinuities, we superimpose the ground-truth motif residues and the generated motif segments before replacement. With no fine-tuning and no conditioning, PAR generates plausible scaffolds that preserve motif structures with high fidelity. This stands in contrast to diffusion or flow-based frameworks, which typically require fine-tuning on additional conditions such as masks or motif coordinates, or rely on decomposition strategies (Geffner et al., 2025; Watson et al., 2023; Wang et al., 2023). Moreover, the generated scaffolds differ substantially from the input structure, showing that PAR generates structurally diverse scaffolds rather than merely copying. For example, the leftmost example in Fig. 4 preserves the yellow motif helix while introducing new secondary structure elements like $\beta$-sheet and loops, in contrast to the original helices. We further benchmark zero-shot motif scaffolding in Tab. 11 in the appendix, following evaluation protocols in (Geffner et al., 2025; Lin et al., 2024).

### 4.3. Empirical Analysis of Multiscale PAR

**Scaling effects of PAR.** We examine the model's behaviors by varying the backbone decoder's size and number of training steps in Fig. 5. We train PAR with 3 scales over three different model sizes with 60,200,400 million parameters and three training durations over 200, 400, 600K steps. PAR demonstrates favorable behavior when scaling both model size and training duration, effectively improving its ability to capture the protein data distribution with FPSD scores of 187 against PDB and 170 against AFDB (first two columns in Fig. 5). Further, the fS scores, which reflect quality and diversity, increase with larger model sizes and greater computational budgets. While extending training duration alone offers negligible gains, increasing model size substantially enhances designability, leading to lower sc-RMSD values. Meanwhile, we empirically observe that a moderately sized autoregressive transformer (60M) is sufficient to achieve strong evaluation results. This allows us to reduce computational costs and prioritize increasing the backbone decoder's model capacity that effectively improves generation quality. We provide more discussion on varying model sizes in §C.7.

**Efficient sampling with multi-scale orchestration of SDE/ODE.** While Tab. 1 reports results using a uniform number of sampling steps across scales, the multi-scale formulation of PAR actually offers advantages in sampling efficiency, as shown in Tab. 2. More specifically, (1) sam-

*Table 2.* **Sampling efficiency.** Combining SDE and ODE sampling across scales yields a 2.5× inference speedup compared to the single-scale 400-step baseline, shown in the first and the last row. We generate 100 samples at each length.

| Sampling | Steps | Length 150 | | Length 200 | |
|---|---|---|---|---|---|
| | | Time (s) | Design. (%) | Time (s) | Design. (%) |
| Proteina (SDE) | 0/0/400 | 131 | 97% | 170 | 92% |
| | 0/0/200 | 67 | 89% | 86 | 80% |
| All SDE | 400/400/400 | 312 | 97% | 351 | 94% |
| | 400/400/2 | 184 | 0% | - | - |
| All ODE | 400/400/400 | 312 | 28% | - | - |
| S/S/O | 400/400/400 | 312 | 98% | - | - |
| | 400/400/2 | 184 | 99% | 186 | 91% |
| S/O/O | 400/400/400 | 312 | 96% | - | - |
| | 400/2/2 | **67** | 97% | **68** | 94% |

pling at the coarser scale (e.g., first scale) is more efficient than sampling at finer scales (e.g., 2nd scale) due to shorter sequence length; (2) we can use less number of sampling steps at finer scales than coarser scales. As shown in Tab. 2, by using SDE sampling only at the first scale, and switching to ODE sampling for the remaining scales, PAR could dramatically reduce the diffusion steps from 400 to 2 steps at the last two scales without harming designability (97%), yielding a 4.7x inference speedup. This is possible because a high-quality coarse topology places the model near high-density regions, enabling efficient refinement with ODE sampling. Naively reducing the SDE sampling steps significantly harms designability, dropping to 22% when reducing steps to 50, as shown in Fig. 7. This is consistent with the observation of single-scale models like Proteina, where designability degrades to 89% when reducing SDE sampling steps to 200 in Tab. 2. Crucially, SDE sampling at the first scale is necessary for establishing a reliable global topology, given that ODE-only sampling exhibits poor designability. In addition, we select Proteina as the single-scale baseline as it shares our transformer architecture and PAR reduces to Proteina under a single-scale setting, allowing us to examine the speedup from multi-scale formulation. Compared to the Proteina 400-step baseline, PAR achieves 1.96x and 2.5x sampling speedup at length 150 and 200, respectively. This improvement is driven by speeding up the final scales, where the longer sequence lengths cause computational costs to grow quadratically in transformer architectures. Moreover, the computational costs remain constant at the first scale because it has a fixed size 64, even when generating longer sequences. We detail inference configurations in Tab. 8.

*Table 3.* **Mitigating exposure bias for PAR.** We adopted various training strategies to mitigate the exposure bias for multi-scale autoregressive modeling. These techniques are consistently effective effective in improving structure quality. NCL: Noisy Context Learning. SS: Schedule Sampling.

| Method | sc-RMSD ↓ | FPSD vs. (PDB/AFDB) ↓ | fS-(C/A/T) ↑ |
|---|---|---|---|
| Teacher Forcing | 2.20 | 99.66 / 37.64 | 2.53 / 5.56 / 29.67 |
| + NCL | 1.58 | 89.70 / 23.69 | 2.54 / 5.85 / 28.37 |
| + NCL & SS | 1.48 | 90.66 / 24.59 | 2.54 / 5.84 / 28.77 |

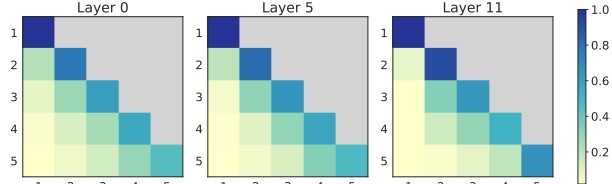

*Figure 6.* Visualization of the average attention scores in PAR autoregressive transformer over 5 scales, obtained from samples with lengths in (128, 256). We provide attention map visualization for shorter proteins in §C.9

**Mitigating exposure bias.** To mitigate exposure bias, we adopted noisy context learning (NCL) and scheduled sampling (SS) as defined in §3.3. Noisy context learning encourages the model to infer structural guidance from corrupted context and boosts the structure generation quality. Tab. 3 shows that noisy context learning effectively improves the sc-RMSD of the generated structure from 2.20 to 1.58, and reduces FPSD against AFDB to 23.69 when using ODE sampling. The designability further improved to 1.48 along with scheduled sampling, which makes the training process more aligned with the inference scenario. Results are obtained with 60M PAR trained for 100K steps.

**Interpreting multi-scale PAR.** We visualize the attention maps of the autoregressive transformer at each scale (Fig. 6). We average the attention scores within each scale, normalize them such that the scores across scales sum to 1, and average them over 50 test samples to obtain the scale-level attention distribution during inference. We summarize three key observations: *(i)* Most scales barely attend to the first scale, since the input to this scale, a `bos` token, carries little structural signal. *(ii)* Each scale primarily attends to the previous scale, which typically contains richer contextual and structural information. *(iii)* Despite focusing most heavily on the current scale, the model still retains non-negligible attention to earlier scales. This indicates that PAR effectively integrates information across multiple scales and maintains structural consistency during generation. This aligns with results in Tab. 5 where the autoregressive Transformer effectively improves consistency with the given prompt. We also observe similar patterns on shorter proteins, as shown by the attention maps in Fig. 10 in the appendix.

*Table 4.* **Multi-scale formulation.** We ablate different strategies for scale configuration in downsampling.

| Define scale | Designability | | FPSD vs. | | fS |
|---|---|---|---|---|---|
| | (%)↑ | (sc-RMSD)↓ | PDB↓ | AFDB↓ | (C / A / T)↑ |
| {64, 256} | 83.0 | 1.38 | 282.85 | 274.32 | 2.14/6.58/20.66 |
| {64, 128, 256} | 85.0 | 1.39 | 279.63 | 267.35 | 2.15/6.52/20.35 |
| {64, 128, 192, 256} | 77.8 | 1.55 | 296.70 | 282.69 | 2.05/6.04/18.69 |
| {64, 96, 128, 192, 256} | 81.0 | 1.51 | 276.00 | 263.58 | 2.17/6.31/20.65 |
| {$L/4, L/2, L$} | 86.4 | 1.49 | 310.64 | 298.30 | 2.00/5.87/18.91 |

**Multi-scale formulation.** We ablate the effect of defining scale *by length* versus *ratio*, as shown in §3.1. Tab. 4

shows that under comparable levels of upsampling ratio ($\{64, 128, 256\}$ and $\{L/4, L/2, L\}$), the *by-length* strategy outperforms *by-ratio*. Meanwhile, PAR obtains better designability and FPSD when increasing from two scales to three scales. Beyond this point, increasing the scale configurations to four and five scales results in degraded designability, potentially due to error accumulation and exposure bias. These results support our choice of adopting the 3-scale PAR as the default. All results are obtained using the 60M model.

**AR Transformer improves structural consistency.** We conduct an ablation study to evaluate the effectiveness of the autoregressive transformer in Tab. 5. We compare two different strategies for encoding prior-scale structural context, including *(i)* direct input, where the multi-scale structures are directly fed into the backbone decoder without any intermediate encoding; and *(ii)* transformer encoder, where all scales are processed autoregressively by a Transformer encoder, and the resulting encoded representation is then passed to the backbone decoder. We train two 60M models and evaluate both models by downsampling 588 testing structures as prompts and re-upsamples them with PAR. As shown in Tab. 5, the transformer encoder demonstrates better structural consistency, indicating that autoregressive encoding across scales produces coherent structural guidance over scales, consistent with attention maps in Fig. 6.

*Table 5.* **Structural consistency for prompted generation.** Using a transformer to encode prior-scale structural conditions shows better prompt-following than direct input.

| | RMSD vs. Reference ↓ | | | TM-score vs. Prompt ↑ | | |
|---|---|---|---|---|---|---|
| *Length* | (32.64] | (64,128] | (128,256] | (32.64] | (64,128] | (128,256] |
| *Reference* | - | - | - | 0.60 | 0.61 | 0.59 |
| Direct Input | 2.13 | 3.38 | 6.51 | 0.58 | 0.61 | 0.59 |
| Trans. Encode | 1.45 | 2.72 | 5.75 | 0.60 | 0.64 | 0.61 |

## 5. Discussion

PAR is the first multi-scale autoregressive model for protein backbone generation, offering a general framework that includes flow-based methods as a special case. PAR addressed limitations of standard autoregressive models, such as unidirectional dependency, discretization, and exposure bias. Our method robustly models structures over multiple granularities and in turn enables strong zero-shot generalization. This capability includes coarse-prompted conditional generation using points (*e.g.*, 16 points) as structural layout and finer-grained controls such as atomic-coordinate-based motif scaffolding. For unconditional backbone generation, PAR exhibits powerful distributional fidelity and generation quality. The multi-scale orchestration of SDE and ODE sampling enables efficient sampling at finer scales. The analysis of scale-level attention map provides additional insights into how the multi-scale formulation operates.

We hope that PAR unlocks the potential of autoregressive modeling for protein design. Some promising open directions include: (1) *Conformational dynamics modeling.* PAR can, in principle, perform zero-shot modeling of conformational distributions: we downsample a structure and upsample it with PAR to mimic local molecular dynamics. We leave this exciting application for future research. (2) *All-atom modeling.* This work focuses on backbone $C\alpha$ atoms to prioritize autoregressive design, but it's natural to extend to full-atom representations (Qu et al., 2025). The multi-scale framework offers an advantage for flexible zero-shot prompt-based all-atom designs. We provided more discussion on future work in the appendix.

## Acknowledgments

We thank Dr. Hang Li, Liang Hong, Xinyou Wang, Jiasheng Ye, Yi Zhou, Jing Yuan, Yilai Li, Zhenghua Wang, Yuning Shen, Huizhuo Yuan, as well as other colleagues at ByteDance Seed for their valuable comments and support.

## Impact Statement

Protein design holds significant potential in drug development, vaccine and antibody discovery, industrial biotechnology, and sustainable chemistry. Generative models provide new opportunities to accelerate discovery and deepen our understanding of protein structures, which may bring positive impact to medicine, materials science, and manufacturing.

However, we also acknowledge the potential risks of generative models. To mitigate such risks, this study is conducted solely on publicly available datasets and strictly adheres to relevant ethical guidelines. We advocate for the responsible research and application of protein generative models to ensure that their development truly benefits society.

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

# A. Implementation and Evaluation Details

We follow the implementation of Proteina (Geffner et al., 2025) for training PAR, using the same architecture and hyperparameter setup. Training is conducted on 8 H100 GPUs, with a batch size of 15 per GPU, for a total of 200k steps. We train the flow-based backbone decoder with 60 M, 200 M, and 400 M parameters, using the same non-equivariant transformer architecture as Proteina. For the autoregressive module, we adopt Proteina's smallest configuration (60 M parameters), as we find that a small AR module is enough to yield competitive generation quality, discussed in §4.3. For a fair comparison, we trained Proteina from scratch under the same setting and achieved similar or even better performance than results reported in the original paper. For other baselines, we directly obtain the results from (Geffner et al., 2025). Model and training configurations can be found in Tab. 6. Note that we remove pair representations, triangle update as well as auxiliary loss for memory and training efficiency, and the additional trainable parameters come from the 60M autoregressive transformer encoder.

*Table 6.* **Hyperparameters for PAR models.** $\mathcal{T}_\theta$: autoregressive transformer; $\mathbf{v}_\theta$: flow-based atomic decoder.

| | $\mathcal{T}_\theta$ | $\mathbf{v}_\theta$ | | |
|---|---|---|---|---|
| **PAR Architecture** | 60M | 60M | 200M | 400M |
| initialization | random | random | random | random |
| sequence repr dim | 512 | 512 | 768 | 1024 |
| sequence cond dim | 128 | 128 | 512 | 512 |
| $t$ sinusoidal enc dim | 196 | 196 | 256 | 256 |
| interpolated position enc dim | 196 | 196 | 128 | 128 |
| # attention heads | 12 | 12 | 12 | 16 |
| # transformer layers | 12 | 12 | 15 | 18 |
| # trainable parameters | 60M | 60M | 200M | 400M |

## A.1. Implementation Details

In §3.2 we briefly introduce two novel techniques for our autoregressive modeling: scale embedding and interpolated position embedding.

**Scale Embedding.** Since we use a shared decoder to train across all scales, we introduce a scale embedding to distinguish data distributions at different scales. Each scale is assigned a unique scale id, which is incorporated into the model to help disambiguate the varying statistical characteristics associated with different scales.

**Interpolated Position Embedding.** Interpolated position embedding is a natural extension to the standard position embedding for sequence representation. In the raw structure, each residue is associated with a 3D coordinate and a position ID ranging from 1 to $L$, where $L$ is the protein length. Our downsampled structure and interpolated position embeddings are derived from the raw structure and position IDs via interpolation, following the sequential order of residues. Each interpolated residue is computed by interpolating the coordinates of neighboring real residues, while each interpolated position ID is obtained by interpolating over the corresponding relative positions. This approach has the advantage that, across inputs of different lengths (i.e., different scales), the interpolated positions still reflect the relative location of each interpolated residue within the original structure, providing a coarse-grained view of the real protein.

## A.2. Evaluation Metrics

We evaluate the model from multiple perspectives, including quality and diversity, following evaluation protocols established in prior literature by (Yim et al., 2023c; Bose et al., 2024). Specifically, we sample 100 structures for each of the five sequence lengths: 50, 100, 150, 200, and 250, resulting in a total of 500 structures for evaluation.

**Designability.** Following the procedure from Yim et al. (2023c), we generate 8 candidate sequences for each structure using ProteinMPNN (Dauparas et al., 2022) with a temperature of 0.1. Each sequence is folded into a predicted structure using ESMFold (Lin et al., 2023). We compute the root-mean-square deviation (RMSD) between each predicted structure and the original generated structure, and record the minimum RMSD across the 8 predictions. A structure is considered designable if its minimum RMSD is less than 2 Å. We report the proportion of designable structures and the average minimum RMSD across all samples.

**Diversity.** Following Bose et al. (2024), we compute the average pairwise TM-score among all designable structures for

each sequence length. The final diversity score is obtained by averaging these values across all five lengths.

**Secondary Structure.** To analyze secondary structure characteristics, we annotate all designable structures using the P-SEA algorithm (Labesse et al., 1997) as implemented in Biotite (Kunzmann & Hamacher, 2018). For each structure, we compute the proportion of alpha helices and beta sheets, and report the average proportions across all samples.

To better assess the model's overall structural fidelity at the distributional level, we adopt two metrics introduced in Geffner et al. (2025). We randomly sample 125 structures at each sequence length from 60 to 255 (with a step size of 5), resulting in 5,000 structures in total. Importantly, no designability filtering is applied during this stage, and all samples are used for evaluation.

**Fréchet Protein Structure Distance (FPSD).** Analogous to the Fréchet Inception Distance (FID) (Heusel et al., 2017), FPSD measures the Wasserstein distance between the distributions of generated and reference structures. Structures are embedded into a feature space defined by a fold class predictor, and the distance is computed based on the resulting Gaussian approximations.

**Protein Fold Score (fS).** Inspired by the Inception Score (IS) (Salimans et al., 2016), the fS metric encourages both diversity and sample-level quality. High-quality generations lead to confident fold class predictions, while diversity is captured by the entropy across the predicted fold distribution.

**Novelty.** We adopt the Foldseek command used in Geffner et al. (2025) to compute the maximum TM-score between each generated structure and the reference dataset. The reference set is Foldseek's precomputed PDB database. The exact Foldseek command used in our experiments is provided below.

```
foldseek easy-search <path_samples> <database_path> <out_file> <path_tmp>
--alignment-type 1 --exhaustive-search
--tmscore-threshold 0.0 --max-seqs 10000000000
--format-output query,target,alntmscore
```

We download the precomputed PDB database (pdb100.tar.gz) directly from https://foldseek.steineggerlab.workers.dev/.

### A.3. Unconditional Backbone Generation

We train 200M and 400M models for Proteina and PAR for 200k steps, using Adam optimizer with learning rate 1e-4, no warmup applied. For evaluation, we sample from Proteina and PAR with the same techniques below. We follow the optimal configuration and sample 400 steps for Proteina. For PAR, we find 1k steps show better results.

**Self conditioning.** Self-conditioning has been widely employed in protein design. During sampling, the model's own previous predictions

$$\hat{\mathbf{x}}(\mathbf{x}_t) = \mathbf{x}_t + (1 - t)\mathbf{v}_t^\theta(\mathbf{x}_t) \tag{7}$$

are fed back as conditions to guide subsequent generation. During training, the model is conditioned on its own predictions with a probability of 50%. Sampling can be performed either with or without self-conditioning.

**Low temperature sampling.** In Eqn. 6, the parameter $\gamma$ is injected to control the scale of noise. When $\gamma = 1$, this SDE yields the same marginals as the ODE defined by flow model. In practice, it is common to use a lower $\gamma < 1$ which empirically improves designability at the cost of diversity. In this paper, we use $\gamma = 0.30$ by default.

**Category of unconditional backbone generation baselines.** We categorize each baseline based on their modeling types and frameworks in the table below.

## B. Datasets

The training data is derived from the curated AFDB representative dataset (denoted as $D_{\text{FS}}$, containing 0.6M structures), as processed by Proteina. This dataset ensures both high quality (pLDDT $> 80$) and structural diversity, with sequence lengths ranging from 32 to 256 residues. We follow (Geffner et al., 2025) and split it by 98:19:1 for training, validation and testing. For PDB finetuning, as the dataset used in (Geffner et al., 2025) is not publicly available, we reproduce their filtering protocol and curate a designable subset of 21K samples from PDB.

*Table 7.* **Category of unconditional backbone generation baselines.**

| Method | Type | Framework |
|--------|------|-----------|
| FrameDiff | Frame | Diffusion |
| RFDiffusion | Frame | Diffusion |
| ESM3 | Token | PLM |
| Genie2 | Ca | Diffusion |
| Proteina | Ca | FM |
| PAR | Ca | PAR |

# C. More Empirical Analysis

## C.1. Efficient Sampling with SDE/ODE Orchestration

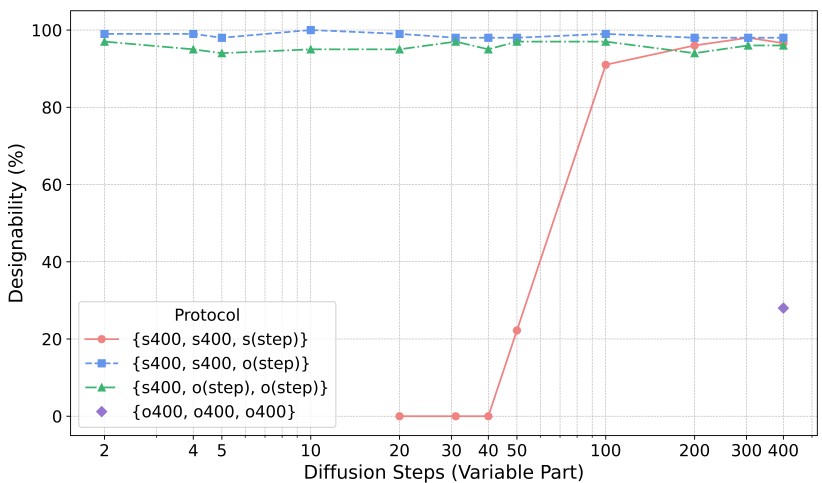

*Figure 7.* **Designability analysis of multi-scale SDE/ODE sampling methods.** Naively reducing the SDE sampling steps substantially degrades the designability (red). Using ODE alone exhibits limited designability (purple). Orchestrating SDE and ODE sampling enables reduced sampling steps while retaining designability (blue and green).

We report the designability over varying sampling steps in Fig. 7. Leveraging SDE sampling at the first scale and ODE for the remaining scales, PAR could effectively reduce diffusion steps without harming designability, highlighting the unique advantage of multi-scale design to orchestrate SDE and ODE sampling at different scales. In addition, aggressively reducing SDE steps or replacing SDE with ODE across all scales yields much worse designability, highlighting the necessity of combining both sampling methods. These results suggest that PAR decomposes backbone generation into coarse topology formation and efficient structure refinement at later scales.

*Table 8.* **Inference configuration and runtime comparison between PAR and Proteina.**

| Setting | PAR | Proteina |
|---------|-----|----------|
| Model size (M) | 60 + 400 | 400 |
| Self conditioning | Yes | Yes |
| Pair representation | No | No |
| Sampling steps | 400 / 2 / 2 | 400 |
| Length | 150 / 200 | 150 / 200 |
| # Samples | 100 | 100 |
| Time (s) | 67 / 68 | 137 / 170 |

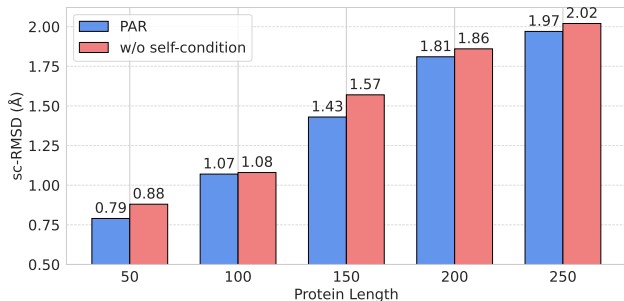

*Figure 8.* **Ablation with self-conditioning.** Self-conditioning consistently improves backbone generation performance of PAR across varying protein lengths, showing that both methods are compatible.

## C.2. Ablation with Self-Conditioning.

Multi-scale autoregressive modeling and self-conditioning similarly guide the generation with a coarse estimate of the structure. To evaluate the role of self-conditioning in our multiscale framework, we conducted an ablation study (Fig. 8), where the results are from the same 60M model in the previous ablation study. Across all length ranges, the model with self-conditioning consistently generates higher-quality protein structures, in terms of sc-RMSD. Although self-conditioning also supplies an intermediate structural estimate during generation, it is complementary to the multi-scale formulation and yields further improvements in structural quality.

## C.3. Long Protein Generation

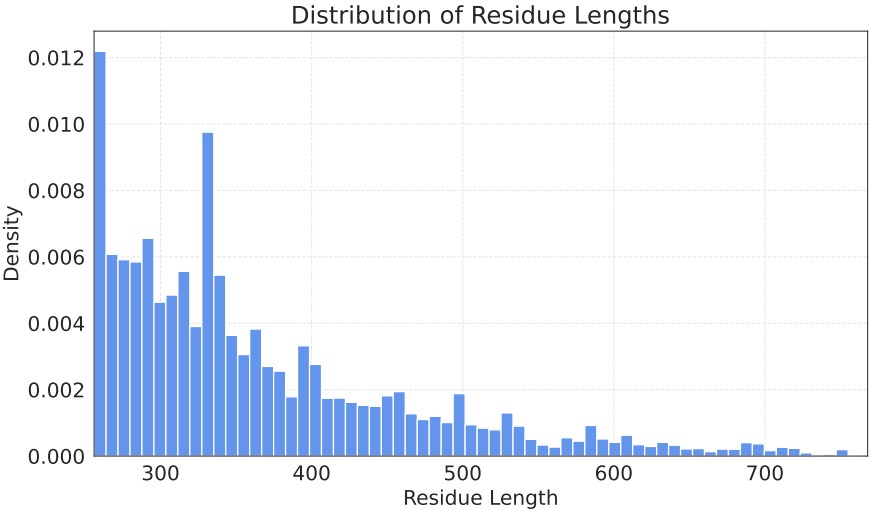

*Figure 9.* Protein length distribution for long protein finetuning.

**Finetuning on longer protein chains.** We follow Proteina to finetune our models on datasets with longer proteins. Since Proteina has not released its long-protein dataset, we cannot fully reproduce their experiment setups. Instead, we follow the filtering procedure described in their appendix on PDB structures to curate a long-protein dataset. We filter PDB structures to lengths between 256 and 768 residues and keep only designable samples, resulting in 26k high-quality proteins. The length-distribution of this dataset (Fig. 9) exhibits a long-tail shape with peaks around 300-400 residues. We then finetune the 400M PAR and Proteina models in Tab. 1 on this dataset for 10k steps.

**Long-protein generation.** We generate 100 proteins for each length in $\{300, 400, 500, 600, 700\}$. PAR exhibits higher designability at lengths $\{300, 400\}$, consistent with the higher density of training samples in this range. At lengths between 500 to 700, both Proteina and PAR show degraded designability, while PAR demonstrating slightly better results. We

attribute this to the long-tail nature of the training set, which includes far fewer samples in the length range between 500 and 700. The limited size of the training set (26K) also potentially hinders the model from reaching its full potential. We leave scaling up long-protein data as a promising direction for future work.

*Table 9.* **Long protein generation.** scR: sc-RMSD (Å) ↓. DesA: Designability (%) ↑.

|  | 300 | | 400 | | 500 | | 600 | | 700 | |
|---|---|---|---|---|---|---|---|---|---|---|
|  | scR | DesA | scR | DesA | scR | DesA | scR | DesA | scR | DesA |
| Proteina | 1.91 | 85 | 2.70 | 61 | 4.09 | 49 | 7.90 | 21 | 13.32 | 4 |
| PAR | **1.28** | **93** | **1.65** | **72** | **3.19** | **52** | **6.80** | **29** | **11.29** | **10** |

## C.4. Foldseek Cluster Diversity

*Table 10.* **Foldseek cluster diversity.**

| $\gamma$ | Designable Clusters |
|---|---|
| 0.35 | 119 |
| 0.40 | 126 |
| 0.45 | 142 |
| 0.50 | 140 |
| 0.60 | 164 |
| 0.70 | 160 |
| 0.80 | 146 |

We investigated the foldseek cluster diversity of PAR-generated samples. A larger $\gamma$ increases sampling stochasticity and improves the diversity, reaching its peak value at $\gamma$=0.6. We generate 500 structures, with 100 samples for each length in $\{50, 100, 150, 200, 250\}$. We use the same foldseek command following Geffner et al. (2025) with a tmscore threshold of 0.5. The command is

```
foldseek easy-cluster <path_samples> <path_tmp>/res <path_tmp>
--alignment-type 1 --cov-mode 0 --min-seq-id 0
--tmscore-threshold 0.5
```

## C.5. Zero-shot Motif Scaffold Benchmark

We quantify the **zero-shot** motif scaffolding performance of PAR in Tab. 11. For other training-based methods, we directly quote the results reported in Proteina (Geffner et al., 2025).

We use PAR to generate 1000 backbone structures for each benchmark problem in Watson et al. (2023). Following Proteina's evaluation protocol, we produce 8 ProteinMPNN sequences with the motif residues fixed, and feed each sequence to ESMFold. Using the predicted structure, we calculate ca-RMSD and MotifRMSD. A design is considered a success if any sequence achieves scRMSD $\leq$ 2Å, a motifRMSD $\leq$ 1Å, pLDDT $\geq$ 70, and pAE $\leq$ 5. Note that our method is the only one evaluated in a *zero-shot* setting, whereas all other baselines rely on training or finetuning with additional motif conditioning.

## C.6. Scale-Agnostic Inference

In our original setup, we included a learnable scale embedding vector as part of the AR module's conditioning. This embedding allows the model to identify the current scale and adjust its behavior (e.g., generating coarse vs. fine structures). However, since the dimensionality of this learnable embedding is fixed to the number of scales, the model cannot be applied to a different scale configuration at inference.

To explore flexible scale configurations, we finetune an alternative model that simply discards the learnable embedding on the PDB designable subset for 5k steps. This formulation cancels the embedding from a fixed number of scales and enables inference across arbitrary scale settings. As shown in the Tab. 12, when inferring with five scales using this 3-scale model, FPSD remains stable, suggesting that the model still captures the underlying data distribution under altered scale

*Table 11.* **Zero-shot motif scaffold benchmark.** PAR* indicates our zero-shot model, producing 1000 samples, while other baselines *require finetuning*. Baseline results are taken directly from Geffner et al. (2025), which reports results using 1000 samples. SR: success rate.

| | Unique Solutions (%) | | | | | Novelty (↓) |
|---|---|---|---|---|---|---|
| | PAR* | Proteina | Genie2 | RFDiffusion | FrameFlow | PAR* |
| 1PRW | 0.1 | 0.3 | 0.2 | 0.1 | 0.3 | 0.785 |
| 1BCF | 0 | 0.1 | 0.1 | 0.1 | 0.1 | - |
| 5TPN | 0 | 0.4 | 0.8 | 0.5 | 0.6 | - |
| 5IUS | 0 | 0.1 | 0.1 | 0.1 | 0 | - |
| 3IXT | 5.9 | 0.8 | 1.4 | 0.3 | 0.8 | 0.794 |
| 5YUI | 0 | 0.5 | 0.3 | 0.1 | 0.1 | - |
| 1QJG | 2.1 | 0.3 | 0.5 | 0.1 | 1.8 | 0.879 |
| 1YCR | 7.2 | 24.9 | 13.4 | 0.7 | 14.9 | 0.854 |
| 2KL8 | 0.1 | 0.1 | 0.1 | 0.1 | 0.1 | 0.859 |
| 7MRX.60 | 0.2 | 0.2 | 0.5 | 0.1 | 0.1 | 0.607 |
| 7MRX.85 | 0.4 | 3.1 | 2.3 | 1.3 | 2.2 | 0.846 |
| 7MRX.128 | 0.4 | 5.1 | 2.7 | 6.6 | 3.5 | 0.879 |
| 4JHW | 0 | 0 | 0 | 0 | 0 | - |
| 4ZYP | 0 | 1.1 | 0.3 | 0.6 | 0.4 | - |
| 5WN9 | 0.1 | 0.2 | 0.1 | 0 | 0.3 | 0.522 |
| 5TRV_short | 0.1 | 0.1 | 0.3 | 0.1 | 0.1 | 0.810 |
| 5TRV_med | 0.4 | 2.2 | 2.3 | 1.0 | 2.1 | 0.858 |
| 5TRV_long | 0.2 | 17.9 | 9.7 | 2.3 | 7.7 | 0.777 |
| 6E6R_short | 1.9 | 5.6 | 2.6 | 2.3 | 2.5 | 0.810 |
| 6E6R_med | 5.8 | 41.7 | 27.2 | 15.1 | 9.9 | 0.848 |
| 6E6R_long | 3.6 | 71.3 | 41.5 | 38.1 | 11.0 | 0.855 |
| 6EXZ_short | 3.0 | 0.3 | 0.2 | 0.1 | 0.3 | 0.811 |
| 6EXZ_med | 4.2 | 4.3 | 5.4 | 2.5 | 11.0 | 0.855 |
| 6EXZ_long | 3.6 | 29.0 | 32.6 | 16.7 | 40.3 | 0.843 |

configurations. However, the designability substantially drops, indicating that sampling with an unseen scale configuration fails to preserve structural detail, ultimately leading to lower-quality results.

*Table 12.* Inference with flexible scale configuration.

| | Designability | | FPSD ↓ | | fS ↑ |
|---|---|---|---|---|---|
| | (%) ↑ | (sc-RMSD) ↓ | vs. PDB | vs. AFDB | (C/A/T) |
| PAR (3 scale) | 96.6 | 1.04 | 160.99 | 228.44 | 2.57/7.42/23.61 |
| w/o scale emb | 92.8 | 1.16 | 175.09 | 246.34 | 2.54/7.66/26.68 |
| 5 scale inference | 72.6 | 1.74 | 177.01 | 246.76 | 2.56/7.53/26.78 |

## C.7. Ablating AR and Decoder Size

*Table 13.* **Effect of AR module and decoder size.** Both AR and decoder utilize transformer-based architectures.

| AR | Decoder | sc-RMSD | Designability (%) |
|---|---|---|---|
| 400M | 60M | 1.26 | 87.80 |
| 60M | 400M | 1.01 | 96.00 |
| 60M | 60M | 1.19 | 92.60 |

We introduced an ablation study examining the AR encoder size, and discussed crucial design choices for both the AR encoder and flow-based decoder. We summarize key findings below.

**Per-token vs per-scale decoder**. In our preliminary study, we implemented the model with a 200M-parameter AR module and, following MAR (Li et al., 2024), used a 3-layer MLP ( 20M) as the diffusion head. However, this setup failed to generate reasonable structures, yielding an average sc-RMSD of 16. This likely occurs because a per-token decoder is not expressive enough to capture the global correlations between atoms that is required to produce a reliable coarse structure at the first scale, which is crucial for the subsequent coarse-to-fine refinement. These observations motivated our shift to a

per-scale transformer-based decoder.

**Large vs. small decoder.** As shown in Tab. 13 and our scaling experiments in §4.3, using a large decoder brings effective improvements to generation quality.

**Large AR vs small AR.** With the decoder size fixed, increasing the AR transformer size from 60M to 400M does not offer improvements. One one hand, this is consistent with the module's role to generate scale-wise conditioning to guide the backbone generation, which does not require large model capacity—a similar trend observed in image generation (Chen et al., 2025). In addition, we believe this is due to exposure bias: the AR module overfits to ground truth context to stabilize training, resulting in a mismatch with inference, where the model relies on its predictions as context. This issue becomes more severe under several conditions:

(1) Larger AR models tend to overfit the context more strongly, making exposure bias more severe.

(2) Limited data increases overfitting risks: our 588K training structures (32–256 residues each) provide far less coverage than datasets like ImageNet (1.28M 256x256 images).

(3) High precision tasks like protein modeling are sensitive to small errors, making exposure bias more serious than in image generation, where the compressed VAE latents lie in a smoother Gaussian space that is robust to small errors at the cost of some visual details (Zheng et al., 2026; Li & He, 2025).

Our noisy context learning and scheduled sampling mitigate this issue for the 60M PAR, but scaling the AR transformer appears to intensify this issue. Exploring more training data is a potential solution and we leave this for future work.

### C.8. Sequence-Based Downsampling Preserves Pairwise Spatial Relationships

*Table 14.* RMSE and LDDT across different downsample sizes.

| Size(i) | 16 | 32 | 64 | 128 |
|---------|------|------|------|------|
| **RMSE** | 0.362 | 0.275 | 0.217 | 0.170 |
| **LDDT** | 1 | 1 | 1 | 1 |

We discuss whether 1D downsampling properly preserves pairwise spatial relationships. To study this, we attempt to investigate the difference between pairwise distances computed after downsampling the 1D coordinate sequence and those obtained by downsampling the full-resolution 2D distance map. We discuss details below.

**Spatial relationships in downsampled 1D sequence.** We follow the process below to quantify the spatial relationships:

1. Downsample the coordinate sequence from $\mathbb{R}^{L \times 3}$ to $\mathbb{R}^{size(i) \times 3}$ for each scale i.

2. We compute pairwise distance maps using the downsampled sequence, leading to a `size(i)` $\times$ `size(i)` map.

**Spatial relationships in 3D space after downsampling.** We quantify this using the pairwise distance map calculated from the full-resolution structure:

1. Calculate the pairwise distance map of the structure, producing a $L \times L$ map.

2. We downsample pairwise map this using the `F.interpolate(mode='bicubic')` operation, resulting in a `size(i)` $\times$ `size(i)` map.

**Does sequence-based downsampling preserve spatial relationships?**

We select all samples from the testing set, and calculate the RMSE and LDDT between the aforementioned two `size(i)` $\times$ `size(i)` pairwise maps for each sample. As expected, rmse slightly increases as `size(i)` decreases, reflecting the loss of fine-grained details at coarser scales. However, lddt remains consistently at 1 and the rmse values remain low across all scales. Together, these results indicate that, despiste small information loss at the coarse scales, 1D sequence downsampling preserves the essential pairwise spatial correlations captured by the downsampled 2D distance map.

## C.9. Visualization of Attention Scores

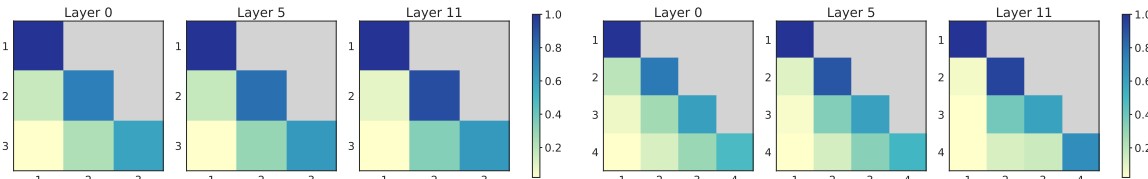

*Figure 10.* Visualization of the average attention scores in PAR autoregressive transformer over 3/4 scales. Left Length ∈ (32, 64]. Right Length ∈ (64, 128].

We provide attention score visualization for shorter proteins in Fig. 10. The pattern generally aligns with Fig. 6, where each scale primarily attends to its previous scale.

## C.10. Training Time Comparison

We report training time for 100k steps on 8 H100s (batch 15, diffusion multiplicity 2).

*Table 15.* Training time.

| Model | Training Time |
|---|---|
| **Proteina (400M)** | 18 hrs |
| **PAR (60+400M)** | 23 hrs |

# D. Other Related Work

**Flow and diffusion-based structure generative models.** Flow-based and diffusion methods have been widely applied to protein backbone generation, with examples including RFDiffusion (Watson et al., 2023) and Chroma (Ingraham et al., 2023). Subsequently, various protein representations have been proposed for protein structure generation. FrameDiff, FoldFlow and FrameFlow (Yim et al., 2023b; Bose et al., 2024; Yim et al., 2023a) model protein structures through per-residue rotation and translation predictions, employing a frame-based Riemannian manifold representation (Jumper et al., 2021; Huang et al., 2022). Building upon FrameFlow, Multiflow (Campbell et al., 2024) jointly models sequence and structures. In contrast, Genie and Genie2 (Lin & AlQuraishi, 2023; Lin et al., 2024) generate protein backbones by diffusing the Cα coordinates. Pallatom and Protpardelle (Qu et al., 2025; Chu et al., 2024) further generate fully atomistic proteins that include side-chains. Meanwhile, Proteina (Geffner et al., 2025) leverages a non-equivariant transformer architecture to model the Cα backbone coordinates, exhibiting scalability and simplicity. In addition to continuous diffusion and flow-matching based approaches, discrete diffusion methods like ESM3 (Hayes et al., 2025) and DPLM-2 (Wang et al., 2025) have been trained on structure tokens, which often reduce structural fidelity and thus limit structure generation quality (Hsieh et al., 2025).

# E. The Use of Large Language Models

We employ large language models exclusively for language-editing, which is limited to polishing text to improve readability. No language models contributed to the development of research ideas, analysis, model, or interpretation of results.

# F. Discussion on Future Work

We outline feasible future directions to extend PAR for all-atom design, and modeling of other biomolecules.

**All-atom design.** Following a P(all-atom)-style design (Qu et al., 2025), one natural extension is to add two finer scales on top of the current framework: a backbone-atom scale and an all-atom scale. Concretely, the model could generate backbone atoms (N, Ca, C) conditioned on Ca and previous coarse scales, and then generate full side-chain atoms (e.g., atom14 representation) conditioned on the backbone atoms. An additional output head for residue type prediction could also be applied. A more modular alternative is to leverage Full-atom MPNN (FAMPNN) (Widatalla et al., 2025), which jointly predicts sequence and all-atom structure from a backbone input produced by PAR.

**Molecular dynamics.** With the multi-scale formulation, PAR might be extended for producing protein ensembles by 1) training with ensemble data and being provided with sequence condtioning; and 2) leveraging coarse-to-fine modeling by upsampling the same coarse-grained structure, which is a one-to-many mapping, to produce a new protein ensemble.

**Multi-chain and other biomolecular modalities.** For multi-chain generation, one natural direction is to follow AlphaFold-style designs and introduce chain identifiers to distinguish different biomolecules while modeling their joint structure. For other biomolecules, the key requirement is again a suitable coarse-grained representation: for linear polymers such as DNA/RNA, a protein-like interpolation is likely feasible, while for ligands or other non-linear small molecules, fragmentation-based coarse-graining would be a natural choice. We believe the proposed multi-scale formulation is well aligned with interaction modeling and binder design, since these problems also involve structure generation and coordination across multiple spatial scales.

