# OpenReview forum: "Protein Autoregressive Modeling via Multiscale Structure Generation"
_ICML.cc/2026/Conference — ICML 2026 spotlight_

### Official Review · Reviewer_7cWE · 2026-03-02

**Soundness:** 3
**Presentation:** 4
**Significance:** 3
**Originality:** 3
**Overall Recommendation:** 5
**Confidence:** 4

**Summary:**

The authors introduce PAR, a protein-centric autoregressive modeling regime for 3D generative modeling. Via next-scale prediction conditioning, PAR achieves the scaling and zero-shot performance benefits of autoregressive language models while accounting for the inherent bidirectionality of 3D proteins. Numerous ablation studies confirm the importance of each of PAR's design components.

**Compliance With Llm Reviewing Policy:**

Affirmed.

**Final Justification:**

The authors' rebuttals have addressed my initial concerns for this manuscript. Overall, I am looking forward to seeing this work published, as I believe it offers unique ideas and results that may prove useful in other areas of scientific machine learning.

**Key Questions For Authors:**

1. How would the quadratic complexity of autoregressive transformers be considered if/when incorporating all-atom context for protein generation? Would there still be speed benefits of multi-scale generation over single-scale flow matching?
2. Does the downsampling procedure the authors propose only work for proteins, or might it be generalizable to ligands, RNA, etc?
3. Might it be possible to further reduce the number of (coarse-scale) flow matching solver steps by leveraging something like Mean Flows [2], or would PAR be more numerically sensitive during training compared to single-scale flow matching models?

**Limitations:**

Yes.

**Strengths And Weaknesses:**

**Points of Strength:**

1. The authors present PAR, a methodologically and empirically compelling method for autoregressively generating protein backbones with strong scaling and zero-shot prediction capabilities.
2. The authors account for the bidirectionality of 3D protein structures via next-scale prediction.
3. Numerous ablation studies confirm the importance of each of PAR's model components.
4. The authors carefully describe the benefits and limitations of their autoregressive formulation to protein structure generation.

**Points for Improvements:**

1. The authors do not discuss whether PAR is as time-efficient to train as Proteina [1]. They train it for the same number of steps as Proteina, but its total training time (compared to that of Proteina) is not listed.
2. Like Proteina, PAR can currently only generate protein structures consisting of only carbon-alpha (Ca) atoms, meaning it cannot be applied to other related tasks involving precise structural interactions with ligands and other biomolecules. All-atom context will be necessary to unlock new application areas (which the authors briefly discuss).
3. Further, PAR cannot generate protein sequences directly, relying on inverse folding methods for sample evaluation (like Proteina).

**References:**

[1] Geffner, T., Didi, K., Zhang, Z., Reidenbach, D., Cao, Z., Yim, J., ... & Kreis, K. (2025). Proteina: Scaling flow-based protein structure generative models. ICLR 2025.

[2] Geng, Z., Deng, M., Bai, X., Kolter, J. Z., & He, K. (2025). Mean flows for one-step generative modeling. NeurIPS 2025.

---

> ### Author Rebuttal · Authors · 2026-03-31
>
> > `W1` The authors do not discuss whether PAR is as time-efficient to train as Proteina [1]. Total training time is not listed.
>
> We report training times for 100k steps on 8 H100s (batch 15, diffusion multiplicity 2). Table will be updated in Sec. C.
>
> | Model                               | Training time |
> |------------------------------------|---------------------------|
> | Proteina 400M | 18 hours     |
> | PAR 60+400M   | 23 hours |
>
> > `W2` Model other biomolecules and all-atom context to unlock new application areas.
>
> Thank you for this important comment. We agree that all-atom context might be helpful for applications involving precise structural interactions, such as ligand binding, binder design, and other multimolecular settings.
>
> While PAR is most naturally suited to chain-like macromolecules, extending it to complexes, nucleic acids, or protein-ligand systems require suitable multi-scale representations and a finer structural representation beyond Ca backbones.
>
> We view this as a promising next step. As briefly discussed in the paper, one direction is to extend PAR toward backbone-atom and all-atom generation in a coarse-to-fine manner; another is to couple PAR with downstream all-atom models that infer sequence and side-chain structure from generated backbones. We thank the reviewer and will clarify it more explicitly in the revision.
>
> > `W3` PAR cannot generate protein sequences directly, relying on inverse folding methods for sample evaluation (like Proteina).
>
> Thank you for this comment. In this work we intentionally focus on multi-scale backbone structure generation as the primary problem setting to validate the multi-scale AR design.
>
> We view sequence generation as an important next step. As discussed above for possible all-atom extensions, one direction is to augment PAR with an additional module for residue-type prediction within the coarse-to-fine generation process; another is to combine PAR with models such as FAMPNN, which can infer sequence (and potentially full-atom structure) from the generated backbone.
>
> Thus, the current formulation is a backbone-centered foundation and can be naturally extended toward joint backbone-sequence generation in future work.
>
> > `Q1` How would the quadratic complexity of autoregressive transformers be considered when incorporating all-atom context? Would there still be speed benefits over single-scale flow matching?
>
> Thank you for this question. If all-atom context is incorporated, we would likely still follow an AlphaFold-style per-residue tokenization, rather than treating every atom as an independent token. In that case, the transformer complexity would remain primarily tied to the number of residues, while all-atom information would be represented within each residue through additional structural channels or finer decoding stages.
>
> The main speed advantage of PAR comes from reducing the number of sampling steps at fine scales, rather than reducing the cost of a single forward pass. The multi-scale formulation allows allocating most sampling budget at coarse scales with shorter sequence lengths, and only a few refinement steps are used at finer scales.
>
> Therefore, we believe the same principle would still provide speed benefits over single-scale flow matching even in a richer all-atom setting, although the exact trade-off would depend on the final architectural choice.
>
> > `Q2:` Does the downsampling procedure generalize to ligands, RNA, etc?
>
> Thank you for this question. We believe the current downsampling procedure is most naturally suited to biomolecules with a *linear backbone*, such as proteins and likely also RNA, where coarse-graining can be defined along the chain while preserving the global geometry.
>
> For ligands or other non-linear small molecules, the same downsampling scheme is not directly applicable, but might be achieved by adopting coarse-graining or fragmentation strategies, such as Murcko scaffolds or principal subgraph-based decompositions.
>
> We view the key requirement as not the specific downsampling rule itself, but the availability of a meaningful coarse-grained representation that preserves the structural dependencies relevant to generation.
>
> > `Q3` Might it be possible to further reduce the number of (coarse-scale) flow matching solver steps by leveraging something like Mean Flows [2], or would PAR be more numerically sensitive during training compared to single-scale flow matching models?
>
> Thank you for this interesting suggestion. We suggest that methods for reducing the number of solver steps, such as Mean Flows and related few-step / one-step flow distillation approaches (e.g. Progressive Distillation), could potentially be combined with PAR to further reduce flow matching steps.
>
> Besides, we do not observe training instability in PAR: in our experiments, the model trains stably with mixed precision training, and the main challenge appears to be preserving coarse-scale generation quality under more aggressive solver reduction.

---

> > ### Author Rebuttal · Reviewer_7cWE · 2026-03-31
> >
> > I'd like to thank the authors for their thoughtful and comprehensive rebuttals. My core concerns have been addressed. Overall, I am comfortable with my initial score, "accept", and am looking forward to seeing this work published.

---

> > > ### Author Response · Authors · 2026-03-31
> > >
> > > Dear Reviewer 7cWE,
> > >
> > > Thank you for your supportive comments. We sincerely appreciate your efforts for the thoughtful discussion and the opportunity to fully address your concerns.
> > >
> > > Best,
> > >
> > > Authors

---

### Official Review · Reviewer_3GBT · 2026-03-06

**Soundness:** 4
**Presentation:** 4
**Significance:** 4
**Originality:** 4
**Overall Recommendation:** 6
**Confidence:** 4

**Summary:**

This work presents protein autoregressive modeling (PAR), an autoregressive protein backbone generator. While contemporary works focus on diffusion modeling for backbone generation, the authors present a compelling argument motivating autoregressive backbone generation, leveraging a flow-based decoder conditioned on learned multi-scale representations to generate realistic structures. The authors cleverly overcome current pitfalls in autoregressive model training, including exposure bias and discretization concerns.

**Compliance With Llm Reviewing Policy:**

Affirmed.

**Final Justification:**

My recommendation for this work is a strong accept (6). This is a very strong technical paper which solves longstanding problems with autoregressive backbone generation in an area which is dominated by diffusion models. I disagree with assessments from other reviewers claiming lack of novelty. While the core method is an extension of VAR, I believe there are significant contributions toward autoregressive modeling which are both effective and novel, including: noisy context learning, scheduled sampling, and the choice of a flow-based decoder over alternative discretization schemes. My previous score of 5 was due to the lack of code availability, which the authors adequately addressed during the rebuttal.

**Key Questions For Authors:**

When trained at PDB scale, PAR shows a significant drop in alpha helix generation; this seems contradictory to what one might expect given prior trends in the table and previous literature. Is there any sense for why this behavior is occurring?

Comments:

It would be interesting for the authors to consider the efficacy of PAR on novel structures, such as those described in “Principles for designing ideal protein structures” (Koga et al., 2012). This seems like a space where AR models clearly have favorable properties over diffusion for backbone generation.

I believe PAR has the potential to change the current paradigm for backbone generation. While I have kept my review (relatively) short, I have done so because there is not much to critique about this work. It is clear that PAR will be a valuable tool for the community.

**Limitations:**

I would be remiss to not at least mention that the paper does not provide an open source implementation of PAR, or at least an interface to work with PAR. While the paper details a reproducible architecture, it is important to share work with the community so that protein design research can be accelerated.

**Strengths And Weaknesses:**

Strengths:

The methodology is strongly motivated by previous work across domains.

The use of Noisy Context Learning and Scheduled Sampling neatly avoids common issues with autoregressive training schemes.

Discussion on scaling and sampling makes PAR an appealing choice given that autoregressive generation can be prohibitively slow.

The authors demonstrate the ability to perform both conditional and unconditional generation, allowing PAR to be a versatile design model.

The choice of a flow-based decoder over standard VQ-VAE discretization is a well motivated design choice. It seems that at multiple points in this work, the authors proactively avoid imprecise design choices that would have otherwise been acceptable, giving PAR good footing for future works.

Overall, the paper is very well written and easy to follow.



Weaknesses:


The authors claim that a strength of PAR is that by avoiding a discretization scheme, they are able to model bidirectional biophysical relationships. However, this is a strong claim that feels unsubstantiated without further analysis. Perhaps the authors could assess the likelihoods of backbone conformers to strengthen this claim?

Section 3.2 could use additional clarification. It is not exactly clear why PAR resolves the issue of long distance contact modeling without addressing shortcomings in positional encoding schemes or attention biases.

---

> ### Author Rebuttal · Authors · 2026-03-31
>
> > I believe PAR has the potential to change the current paradigm for backbone generation. It is clear that PAR will be a valuable tool for the community.
>
> We really appreciate your valuable comments, time, and efforts and address your questions below.
>
> > `W1` The authors claim that a strength of PAR is that by avoiding a discretization scheme, they are able to model bidirectional biophysical relationships. However, this is a strong claim that feels unsubstantiated without further analysis. Perhaps the authors could assess the likelihoods of backbone conformers to strengthen this claim?
>
> Thank you for this helpful comment. We agree that our wording could be clearer here. Our claim is *not* that avoiding discretization enables bidirectional biophysical modeling. These are two parallel aspects of PAR:
> - by operating directly in continuous space, PAR avoids discretization and better preserves structural fidelity;
> - by adopting a coarse-to-fine multi-scale autoregressive formulation, PAR avoids residue-wise unidirectional ordering, since each scale represents the entire chain at a given resolution rather than only a prefix of residues.
> Thus, bidirectional dependency arises from multi-scale generation while discretization is avoided via the continuous flow-based decoder. We thank the reviewer for pointing out this ambiguity and will revise the text.
>
> > `W2` Section 3.2 could use additional clarification. It is not exactly clear why PAR resolves the issue of long distance contact modeling without addressing shortcomings in positional encoding schemes or attention biases.
>
> Thank you for this question. We would like to clarify that PAR does *not* explicitly resolve long-distance contact modeling through modified pair representations or attention biases. In fact, we remove pair representations for efficiency and do not introduce a specialized attention bias. We do, however, use interpolated positional embeddings, which provide residue position information uniformly distributed across the entire chain at each scale.
>
> The main mechanism by which PAR *alleviates* long-range dependency modeling is instead the *coarse-to-fine multi-scale formulation*. Each scale represents the entire chain at a given resolution, so the model conditions on global coarse-grained structure rather than only a residue prefix used in residue-wise autoregression.
>
> We agree that this point could be explained more clearly and will revise the text in Section 3.2.
>
> > `Q1` When trained at PDB scale, PAR shows a significant drop in alpha helix generation; this seems contradictory to what one might expect given prior trends in the table and previous literature. Is there any sense for why this behavior is occurring?
>
> Thank you for raising this question. Our understanding is that the drop in alpha-helix frequency after PDB fine-tuning is mainly a **data-distribution effect**. In particular, the alpha helix ratio in PDB is much lower than AFDB. We will clarify this point in the revision and discuss it in terms of the difference between AFDB and PDB secondary-structure composition.
>
> | Dataset   | Sec. Struct. % (α / β) |
> |-----------|-------------------------|
> | PDB ref.  | 35.0 / 16.8            |
> | AFDB ref. | 44.9 / 12.7            |
>
> (data from Proteina)
>
> > `Comment` It would be interesting for the authors to consider the efficacy of PAR on novel structures, such as those described in “Principles for designing ideal protein structures” (Koga et al., 2012). This seems like a space where AR models clearly have favorable properties over diffusion for backbone generation.
>
> Thank you very much for pointing out this highly relevant paper. We agree that this is a very promising direction. In fact, one strength of PAR is that its multi-scale formulation naturally supports the injection of structural constraints, including valuable domain knowledge and structural priors such as those described in Koga et al.
>
> We have already provided initial evidence of this capability through 0-shot prompt-based generation and 0-shot motif scaffolding, which demonstrate that PAR can accommodate structure-constrained generation without task-specific retraining. We look forward to exploring how the design principles could be incorporated into PAR and thank the reviewer for highlighting this exciting connection.
>
> > `Limitation` I would be remiss to not at least mention that the paper does not provide an open source implementation of PAR, or at least an interface to work with PAR. While the paper details a reproducible architecture, it is important to share work with the community so that protein design research can be accelerated.
>
> Thank you for raising this important point. We fully agree that sharing implementations is crucial and plan to release the PAR codebase upon acceptance. In addition, during the rebuttal period, we can also consider providing an anonymous link with inference code and model checkpoints to facilitate reviewer evaluation and improve reproducibility.

---

> > ### Author Rebuttal · Reviewer_3GBT · 2026-03-31
> >
> > My concerns were adequately addressed.

---

> > > ### Author Response · Authors · 2026-03-31
> > >
> > > Dear Reviewer 3GBT,
> > >
> > > Thank you for taking time to carefully review our work and for reading our rebuttal. We sincerely appreciate your positive comments and insightful feedback, and we are grateful for the opportunity to address your concerns.
> > >
> > > Best,
> > >
> > > Authors

---

### Official Review · Reviewer_7X8S · 2026-03-13

**Soundness:** 3
**Presentation:** 3
**Significance:** 3
**Originality:** 3
**Overall Recommendation:** 4
**Confidence:** 4

**Summary:**

This paper introduces the first next-scale prediction-based protein structure generation model. To overcome the exposure bias issue, the authors further adopt noisy context learning and scheduled sampling to improve it. Extensive evaluations have shown comparable results to recent baselines.

**Compliance With Llm Reviewing Policy:**

Affirmed.

**Final Justification:**

I have raised my score. The rebuttal provides a convincing clarification of the originality, highlighting key nontrivial challenges solved in adapting next-scale prediction to protein design. The explanation of the source of sampling speedup is clear and adequately supported by empirical evidence, addressing my concern regarding inference efficiency.

**Key Questions For Authors:**

See weaknesses 2 and 3.

**Limitations:**

For now, PAR can only be applied to single chain protein generation. To improve the impact of this method, a possible choice is to make PAR applicable to multi-chain generation, binder design, and multi-modality (ligand, DNA and RNA).

**Strengths And Weaknesses:**

Strengths：
1. This paper first introduces next-scale prediction to protein structure generation.
2. It proposes using noisy context learning and scheduled sampling to address exposure bias.
3. It provides extensive evaluations.

Weaknesses：
1. Originality: While next-scale prediction is impressive, this component is known in other domains. The contributions lie in integration and application, but conceptually the work might be seen as an incremental advancement.
2. The authors claim that PAR achieves a 2.5x sampling speedup compared to single-scale baselines and flow-based models are special cases of PAR. Then, where does the speedup come from? In addition, there is a lack of rigorous inference speed comparison with baselines.
3. If scaling the autoregressive transformer has minimal impacts on the evaluation results, then what is the rationale of adopting AR generation?

---

> ### Author Rebuttal · Authors · 2026-03-31
>
> > `W1` **Originality:** While next-scale prediction is impressive, this component is known in other domains. The contributions lie in integration and application, but conceptually the work might be seen as an incremental advancement.
>
> We respectfully believe the originality of our work goes beyond an incremental integration. The main novelty is **a new formulation for protein design**: coarse-to-fine multi-scale generation, which, to our knowledge, is the first to be explicitly proposed and empirically validated for protein backbone design with strong designability.
> Beyond adopting next-scale prediction at a high level, **our work also resolves several nontrivial challenges** that arise specifically in the protein design setting, including but not limited to:
>
> - how to construct coarse-grained protein representations while preserving global geometry (Sec. 3.1)
> - how to generate fine-grained structures with a flow-based decoder rather than direct prediction as in VAR-style models (Sec. 3.2)
> - how to mitigate autoregressive exposure bias through NCL and scheduled sampling (Sec. 3.3).
> - how to represent relative positional relationships across scales via interpolated positional encodings (Sec. 3.2).
>
> Moreover, the gains are not merely incremental in benchmark scores. The proposed formulation also enables **substantially improved sampling efficiency** and supports **structure-constrained 0-shot generation**, which we believe highlights the broader potential of this formulation beyond the current experiments.
>
> > `W2` The authors claim that PAR achieves a 2.5x sampling speedup compared to single-scale baselines and flow-based models are special cases of PAR. Then, where does the speedup come from? In addition, there is a lack of rigorous inference speed comparison with baselines.
>
> Thank you for the question. In Table 2, Section 4.3, and Appendix C.1, we compare the inference speed of our method with Proteina. Proteina is an ideal baseline as it shares our transformer architecture and PAR reduces to Proteina under a single-scale setting, allowing a fair speed comparison.
>
> The speedup comes from the **multi-scale formulation** rather than from flow matching itself. Empirically, we find that PAR can allocate most of the sampling budget to the first coarse scale (e.g., 400 steps at scale 1), and then perform very efficient refinement at finer scales using only 2 steps per scale. In contrast, single-scale baseline requires 400 forward passes at the full resolution.
>
>
>
> > `W3` If scaling the autoregressive transformer has minimal impacts on the evaluation results, then what is the rationale of adopting AR generation?
>
> AR generation has demonstrated many interesting behaviors in our work, such as the zero-shot generation ability. While scaling performance is an important aspect to evaluate, using it as the sole criterion might overlook other merits and emerging properties.
>
> We adopt the AR generation to define a new **coarse-to-fine multi-scale factorization** for protein design, which enables generation to proceed through progressively refined structural scales. This multi-scale generation framework underlies PAR, which is validated by its strong designability. By leveraging fewer sampling steps at finer scales, it improves sampling efficiency, and its ability to capture structural context at multiple granularities supports structure-constrained zero-shot generation, such as generation prompted with coarse layout and motif scaffolding. These properties are all substantiated due to the multi-scale AR formulation.
>
> The scaling results suggest that model capacity is most beneficial in the structure decoder, which does not undermine the rationale and the emerging properties for multi-scale autoregressive generation.
>
>
>
> > `Limitation1` For now, PAR can only be applied to single chain protein generation. To improve the impact of this method, a possible choice is to make PAR applicable to multi-chain generation, binder design, and multi-modality (ligand, DNA and RNA).
>
> Thank you for this valuable suggestion. We would be very interested in extending our multi-scale generation framework to multi-chain and multi-modality settings.
>
> For multi-chain generation, one natural direction is to follow AlphaFold-style designs and introduce chain identifiers to distinguish different biomolecules while modeling their joint structure. For other biomolecules, the key requirement is again a suitable coarse-grained representation: for linear polymers such as DNA/RNA, a protein-like interpolation is likely feasible, while for ligands or other non-linear small molecules, fragmentation-based coarse-graining would be a natural choice.
>
> We believe the proposed multi-scale formulation is well aligned with interaction modeling and binder design, since these problems also involve structure generation and coordination across multiple spatial scales. We thank the reviewer for highlighting these promising future directions.

---

> > ### Author Rebuttal · Reviewer_7X8S · 2026-04-04
> >
> > Thank you for your response. I believe my concerns have been addressed, and I will consider raising my score provided that the content from the rebuttal is incorporated into the final version of the paper.

---

> > > ### Author Response · Authors · 2026-04-04
> > >
> > > Dear Reviewer 7X8S,
> > >
> > > We sincerely thank you for your valuable time and insightful feedback. We greatly appreciate your thoughtful review and the opportunity to improve our manuscript. Based on all comments and discussions, we will incorporate the following revisions into the final version of the paper, including but not limited to:
> > >
> > >
> > > 1. **Novelty Metrics in Table 1**
> > >    We will add novelty numbers to Table 1 to improve the comprehensiveness and clarity of the evaluation.
> > >
> > > 2. **Updated Motif Scaffolding Criteria**
> > >    We will revise the motif scaffolding evaluation by reporting the number of unique solutions and their novelty when generating 1,000 samples.
> > >
> > > 3. **Speed Comparison and Baseline Justification**
> > >    - We will clarify the rationale for selecting Proteina as the baseline in Section 4.3.
> > >    - We will provide detailed inference configurations in Section C.1
> > >
> > > 4. **All-Atom Design, and Multi-Modality**
> > >    Following discussions with you and Reviewer Cjnw, we will add a new Section F outlining feasible future directions for all-atom design, and multi-modal extensions.
> > >
> > > 5. **Rationale for Autoregressive (AR) Generation**
> > >    We will strengthen the motivation and justification for adopting AR generation in Section 4.3
> > >
> > > 6. **Training Time Comparison (PAR vs. Proteina)**
> > >    We will include an additional table in Section C summarizing the training time comparison, as discussed with Reviewer 7cWE.
> > >
> > > We appreciate your constructive suggestions, which have helped us substantially improve the clarity of our work. Feel free to let us know if there are any additional changes you would recommend.
> > >
> > > We sincerely thank you once again.
> > >
> > > Best,
> > >
> > > Authors

---

### Official Review · Reviewer_Cjnw · 2026-03-23

**Soundness:** 2
**Presentation:** 3
**Significance:** 2
**Originality:** 3
**Overall Recommendation:** 5
**Confidence:** 4

**Summary:**

The authors propose PAR, a model that builds on the Proteina architecture and uses an autoregressive transformer to produce conditioning information for a flow-based decoder, which operates on multiple scales. They therefore adopt next-scale instead of next-token generation and evaluate their model on both unconditional and motif-conditioned generation.

**Compliance With Llm Reviewing Policy:**

Affirmed.

**Ethical Review Concerns:**

The authors addressed my concerns wrt baseline comparisons, which is why I increase my score

**Final Justification:**

The authors have addressed some of my concerns regarding the novelty numbers of the method, which is why I increased the score to a weak accept. The concept is interesting, but would benefit from more up to date tasks such as all atom generation etc, that is why I did not increase my score more.

**Key Questions For Authors:**

[Q1] The authors mention speed benefits, but never describe concrete apples-to-apples numbers for their model and baselines: how does PAR compare?

[Q2] The authors mention conformational dynamics as a straightforward extension from their work; however this seems everything but straightforward to me, and the multi-scale generation does not have any impact on the ability to generate conformational examples if the training data is still single state. What do the authors mean by “it can generate ensembles by mimicking molecular dynamics”

[Q3] The authors also mention all-atom design as an easy extension, which could happen in an all-atom way like P(all-atom) or a partially latent one like La-Proteina. Which one do the authors think is the more suited one to their model and why?

**Limitations:**

yes

**Strengths And Weaknesses:**

[S1] Interesting problem formulation and scaling analysis: the authors find a creative way to formulate the backbone design problem and show that scaling time steps as well as model size (especially in the flow-based decoder) helps performance.

[S2] Multiple ablations: the authors ablate and analyse a lot of the components in their architecture, including the use of multiple scales, length generalisation and other tasks and show interesting zero shot capabilities.


[W1] No novelty numbers: this is my biggest criticism; in protein design novelty of the designs is a key concern to judge generalisation quality and performance on unseen tasks like new motifs at inference time. The authors adopt the full evaluation suite from Geffner et al in Proteina, but remove novelty which seems suspicuous. I strongly urge the authors to add novelty to their main table for unconditional benchmark and ideally as well to the motif results, otherwise it could very well be that the model just memorises the training distribution (which does not seem unlikely given the low FPSD scores). Even for motifs where the authors claim that it generalises this is not supported by data; the model might as well just have memorised another protein with a similar motif from the training set and just copy it over, which is impossible to tell without a novelty analysis.

[W2] motif criteria: why evaluate only 100 backbones and not 1000 similar to all baselines? And the criteria of >1% success rate seems rather hand-selected to allow the method to outperform its baselines.

---

> ### Author Rebuttal · Authors · 2026-03-31
>
> > `W1` **Novelty numbers.**
>
> Thank you for raising this point. We agree that novelty is an important criterion in protein design. Since the novelty evaluation workflow is not provided in the original Proteina codebase, we reproduced it following the instructions and the foldseek command in their paper. We report the novelty scores for unconditional generation benchmark below.
>
> | Method | Designability (%) | FPSD | Novelty |
> |---|---:|---:|---:|
> | FrameDiff | 65.4 | 194.2 | 0.82 |
> | RFDiffusion | 94.4 | 253.7 | 0.85 |
> | ESM3 | 22.0 | 933.9 | 0.90 |
> | Genie | 95.2 | 350.0 | 0.80 |
> | Proteina (200M) | 92.8 | 282.3 | 0.85 |
> | Proteina (400M) | 92.6 | 271.3 | 0.84 |
> | PAR (200M) | 87.0 | 252.0 | 0.83 |
> | PAR (400M) | 96.0 | 313.9 | 0.85 |
> | gamma = 0.45 | 88.0 | 231.5 | 0.84 |
> | PAR_pdb | 96.6 | 161.0 | 0.85 |
> | gamma = 0.45 | 88.8 | 176.6 | 0.84 |
>
>
> Our method achieves novelty scores comparable to Proteina and RFDiffusion.
> Importantly, improvements in FPSD do not come at the cost of novelty, indicating PAR captures underlying structural principles rather than reproducing training samples.
>
> > `W2` **Motif Criteria and Novelty**
>
> We report the number of unique solutions and novelty for producing 1000 samples below.
> PAR solved 5 more tasks than with 100 samples. Note that PAR runs zero-shot motif scaffolding without finetuning. We will remove the criteria of ">1% success rate" to improve the clarity.
>
> | task_name     | unique solutions | novelty |
> |---------------|------------------|---------|
> | 1PRW          | 1                | 0.785   |
> | 1BCF          | 0                | -       |
> | 5TPN          | 0                | -       |
> | 5IUS          | 0                | -       |
> | 3IXT          | 59               | 0.794   |
> | 5YUI          | 0                | -       |
> | 1QJG          | 21               | 0.879   |
> | 1YCR          | 72               | 0.854   |
> | 2KL8          | 1                | 0.859   |
> | 7MRX.60       | 2                | 0.607   |
> | 7MRX.85       | 4                | 0.846   |
> | 7MRX.128      | 4                | 0.879   |
> | 4JHW          | 0                | -       |
> | 4ZYP          | 0                | -       |
> | 5WN9          | 1                | 0.522   |
> | 5TRV_short    | 1                | 0.810   |
> | 5TRV_med      | 4                | 0.858   |
> | 5TRV_long     | 2                | 0.777   |
> | 6E6R_short    | 19               | 0.810   |
> | 6E6R_med      | 58               | 0.848   |
> | 6E6R_long     | 36               | 0.855   |
> | 6EXZ_short    | 30               | 0.811   |
> | 6EXZ_med      | 42               | 0.855   |
> | 6EXZ_long     | 36               | 0.843   |
>
> > `Q1` **Apple-to-apple speed comparisons for PAR and the baseline**
>
> In Section 4.3 and Table 2, we compare the inference speed of our method with Proteina. Proteina is an ideal baseline as it shares our transformer architecture and PAR reduces to Proteina under a single-scale setting, allowing a fair speed comparison to examine the speedup from multi-scale formulation and removing other confounding factors like architectures.
>
> The comparison is run on a single H100 GPU. We detail the inference configurations of Table 2 below:
>
> | Setting | PAR | Proteina |
> |---|---:|---:|
> | Model size (M) | 60+400 | 400 |
> | Self conditioning | yes | yes |
> | Pair representation | no | no |
> | Sampling steps | 400/2/2 | 400 |
> | Length | 150/200 | 150/200 |
> | # samples | 100 | 100 |
> | Time (s) | 67/68 | 137/170 |
>
> > `Q2` **What do the authors mean by "it can generate ensembles by mimicking molecular dynamics"**
>
> Thank you for the question. We suggest that PAR might be extended for producing protein ensembles by 1) training with ensemble data and being provided with sequence condtioning; and 2) leveraging coarse-to-fine modeling by upsampling the same coarse-grained structure, which is a one-to-many mapping, to produce a new protein ensemble.
>
> However, we agree that PAR is not intended to model conformational dynamics or MD-like ensembles from single-state data. We will remove this statement in the conclusion section if the reviewer finds it not essential.
>
> > `Q3` **All-atom design extension.**
>
> Thank you for pointing this out. We briefly discuss two feasible directions.
>
> - Following a P(all-atom)-style design, one natural extension is to add two finer scales on top of the current framework: a backbone-atom scale and an all-atom scale. Concretely, the model could generate backbone atoms (N, Ca, C) conditioned on Ca and previous coarse scales, and then generate full side-chain atoms (e.g., atom14 representation) conditioned on the backbone atoms. An additional output head for residue type prediction could also be applied.
> - A more modular alternative is to leverage Full-atom MPNN (FAMPNN) [1], which jointly predicts sequence and all-atom structure from a backbone input produced by PAR.
>
> We thank the reviewer for raising this point, and we would be happy to discuss these extensions further.
>
> [1] Widatalla et al. FAMPNN. ICML 2025.

---

> > ### Author Rebuttal · Reviewer_Cjnw · 2026-04-04
> >
> > My concerns have been addressed, as a result I raise my score

---

> > > ### Author Response · Authors · 2026-04-04
> > >
> > > Dear Reviewer Cjnw,
> > >
> > > We would like to thank you for your thoughtful comments and your valuable time for reviewing our paper. We deeply appreciate the opportunity to respond to your comments and address your concerns.
> > >
> > >
> > > Best,
> > >
> > > Authors

---

### Decision · Program_Chairs · 2026-04-30

**Decision:**

Accept (spotlight)

**Comment:**

Initially, this paper received diverging reviews. Major concerns raised by reviewers include the lack of novelty numbers, unfounded motif criteria, incremental originality, the lack of rigorous comparisons on training time and inference speed, the rationale of adopting AR generation, some claims without sufficient justifications, some unclear descriptions, the lack of code availability, and limited applicability. The rebuttal addressed most of these concerns well, and all reviewers acknowledged that their major concerns were fully resolved. The AC agrees that the problem formulation is interesting, the proposed method is technically sound, and the experimental evaluations are comprehensive. Reviewers did raise valuable concerns that should be addressed. The authors are encouraged to make the necessary changes in the camera-ready version.